Manuscript prepared for Atmos. Meas. Tech.
with version 2014/09/16 7.15 Copernicus papers of the LaTeX class copernicus.cls.
Date: 24 September 2018

# Lidar temperature series in the middle atmosphere as a reference data set. Part A: Improved retrievals and a 20-year cross validation of two co-located French lidars

Robin Wing[1], Alain Hauchecorne[1], Philippe Keckhut[1], Sophie Godin-Beekmann[1], Sergey Khaykin[1], Emily M. McCullough[2], Jean-François Mariscal[1], and Éric d'Almeida[1]

[1]LATMOS/IPSL, UVSQ Université Paris-Saclay, Sorbonne Université, CNRS, Guyancourt, France
[2]Department of Physics and Atmospheric Science, Dalhousie University, Halifax, Canada

*Correspondence to:* Robin Wing (robin.wing@latmos.ipsl.fr)

**Abstract.** The objective of this paper and its companion (Wing et al., 2018b) is to show that ground based lidar temperatures are a stable, accurate and precise dataset for use in validating satellite temperatures at high vertical resolution. Long term lidar observations of the middle atmosphere have been conducted at the Observatoire de Haute-Provence (OHP), located in southern France (43.93° N, 5.71° E), since 1978. Making use of 20 years of high-quality co-located lidar measurements we have shown that lidar temperatures calculated using the Rayleigh technique at 532 nm are statistically identical to lidar temperatures calculated from the non-absorbing 355 nm channel of a Differential Absorption Lidar (DIAL) system. This result is of interest to members of the Network for the Detection of Atmospheric Composition Change (NDACC) ozone lidar community seeking to produce validated temperature products. Additionally, we have addressed previously published concerns of lidar-satellite relative warm bias in comparisons of Upper Mesospheric and Lower Thermospheric (UMLT) temperature profiles. We detail a data treatment algorithm which minimizes known errors due to data selection procedures, a priori choices, and initialization parameters inherent in the lidar retrieval. Our algorithm results in a median cooling of the lidar calculated absolute temperature profile by 20 K at 90 km altitude with respect to the standard OHP NDACC lidar temperature algorithm. The confidence engendered by the long term cross-validation of two independent lidars and the improved lidar temperature dataset is exploited in (Wing et al., 2018b) for use in multi-year satellite validations.

## 1 Introduction

Rayleigh lidar remote sounding of atmospheric density is an important tool for obtaining accurate, high resolution measurements of the atmosphere in regions which are notoriously difficult to measure routinely or precisely. A key strength of this technique is the ability to retrieve an absolute temperature profile from a measured relative density profile with high spatio-temporal resolution, accuracy and precision. This kind of measurement is exactly what is required to detect longterm middle atmospheric temperature trends associated with global climate change and is of great value for routine satellite and model validation (Keckhut et al., 2004).

Comparisons of middle atmospheric temperatures measured from satellites to those measured from lidars have all noted a relative warm bias in lidar temperatures above 70 km. Several recent examples of lidar-satellite relative warm bias in the upper mesosphere can be found in the work of: (Kumar et al., 2003) [5-10 K relative to HALOE]; (Sivakumar et al., 2011) [5-10 K relative to HALOE, 6-10 K relative to COSMIC/CHAMP, 10-16 K relative to SABER]; (Yue et al., 2014) [13 K at 75 km relative to SABER]; (García-Comas et al., 2014) [3-4 K at 60 km relative to SABER and MIPAS]; (Yue et al., 2014) [13 K at 75 km relative to SABER]; (Dou et al., 2009) [4 K at 60 km relative to SABER]; (Remsberg et al., 2008) [5-10 K at 80 km relative to SABER]; and (Taori et al., 2012; Taori et al., 2012) [25 K near 90 km relative to SABER]. The bias is generally attributed to lidar 'initialization uncertainty' and model a priori contributions to the temperature retrieval but, no systematic attempts are made to fully establish this conclusion. These authors also explore the possible influences of tides, lidar-satellite co-incidence criteria, satellite vertical averaging kernels, and satellite temperature accuracy as possible contributing factors.

The work of this paper is to evaluate the suitability of lidars as a reference dataset and to address the problem of systematic errors due to initialization of the lidar algorithm. The subsequent comparison of the improved lidar temperatures to satellite measurements is conducted in the companion paper (Wing et al., 2018b).

This work follows three main goals: i) the introduction of the long term data set and the instrumental changes, ii) treatment of this heterogeneous data set for use in the accompanying paper, and iii) improvement of the temperature algorithm and reduction of the warm bias compared to satellite soundings. These goals cannot be completely separated from each other, but goal i) is broadly addressed in sections 2.1 to 3.2 and 3.4; goal ii) is addressed in sections 3.3 to 3.4 and again in sections 3.5 - 4, and goal iii) is addressed in section 5.

Section 2 of this paper describes the current experimental setup, the specifications of two OHP lidars, and the measurement cadence of two key NDACC (Network for the Detection of Atmospheric Composition Change) lidar systems.

Section 3 of this paper outlines techniques to minimize the magnitude of the aforementioned lidar-satellite temperature bias by systematically detailing a rigorous procedure for the treatment and

selection of raw lidar data and will propose improvements to the standard NDACC lidar temperature algorithm for the UMLT (Upper Mesosphere and Lower Thermospshere) region.

Section 4 of this paper gives the net results of the temperature modifications and system improvements in the LTA lidar at OHP.

Section 5 of this paper compares the lidar temperatures produced by an NDACC certified temperature lidar at 532 nm with temperatures produced by the non-absorbing 355 nm line of a co-located NDACC certified ozone DIAL (DIfferential Absorption Lidar) system. This comparison is conducted using a large database of two co-located lidar systems with the goal of providing confidence in the longterm stability of the lidar technique at both wavelengths. There are currently 10 certified temperature lidars, 6 of which are current in their data submission and have temperature profiles freely accessible online. Similarly, there are 12 certified stratospheric ozone DIAL systems of which 5 systems are current with data submission and are available through the NDACC website. We hope that this work will encourage sites with outstanding data obligations to submit their measurements and for DIAL ozone sites to seek validation for their temperature data products for inclusion in the NDACC database (nda). As an ancillary goal we will show that temperatures produced by the Rayleigh lidar technique are accurate, precise and stable over multiple decades and as such are the ideal type of measurement for use in future ground based validation of satellite temperatures. The result of this demonstration will be used in the companion paper (Wing et al., 2018b) as justification for validating satellite data with lidar temperatures.

## 2   Instrumentation Description

### 2.1   Rayleigh Lidar

The OHP Rayleigh-Mie-Raman lidar, LTA (Lidar Température et Aérosols), uses a seeded Nd:YAG to produce a 532 nm laser source with a maximum power of 24 W. The transmitted beam is passed through a 13X beam expander and has a 30 Hz repetition rate, a 7 ns pulse width, and a beam divergence of less than 0.1 mrad.

The receiver assembly consists of a high and low gain elastic channel for 532 nm, a Mie scatter channel at 532 nm for aerosols, a Raman channel at 607 nm for molecular nitrogen, and a Raman channel at 660 nm for water vapour. A schematic of the telescope array is shown in Fig. 1. The high gain Rayleigh channel consists of four telescopes. At the focal point of each telescope is an actuator-mounted 400 $\mu$m diameter fibre optic. The four fibre optics are bundled to project a single signal onto a Hamamatsu R9880U-110 photomultiplier. The low gain Rayleigh, nitrogen Raman, water vapour Raman and Mie channels all use a single telescope setup and actuator mounted fibre optic. The two Raman channels rely on the largest telescope and the signals are separated by a dichroic mirror. Specifications for each telescope are found in Table 1.

| LTA | Mirror Diameter (cm) | Focal Length (mm) | Field of View (mrad) | Parallax (mm) | Optical Filter Width (nm) | Filter Maximum Transmission (%) |
|---|---|---|---|---|---|---|
| High Gain Rayleigh | 4X 50 | 1500 | 0.27 | 800 | 0.3 | 84 |
| Low Gain Rayleigh | 20 | 600-800 | 1.7 | 257 | 0.3 | 84 |
| Nitrogen Raman | 80 | 2400 | 0.6 | 600 | 1 | $\sim 50$ |
| Water Raman | 80 | 2400 | 0.6 | 600 | 1 | $\sim 50$ |
| Aerosol Mie | 20 | 600-800 | 1.7 | 257 | 0.3 | 84 |

Table 1: Specifications for the LTA receiver assembly.

All channels are sampled using a Licel digital transient recorder with a record time of 0.1 $\mu$s which corresponds to a vertical resolution of 15 m. The high and low gain Raleigh channels are electronically gated at 22 km and 12 km, respectively, to avoid damaging the photomultipliers with large signal returns. Further details can be found in (Keckhut et al., 1993) and (Khaykin et al., 2017).

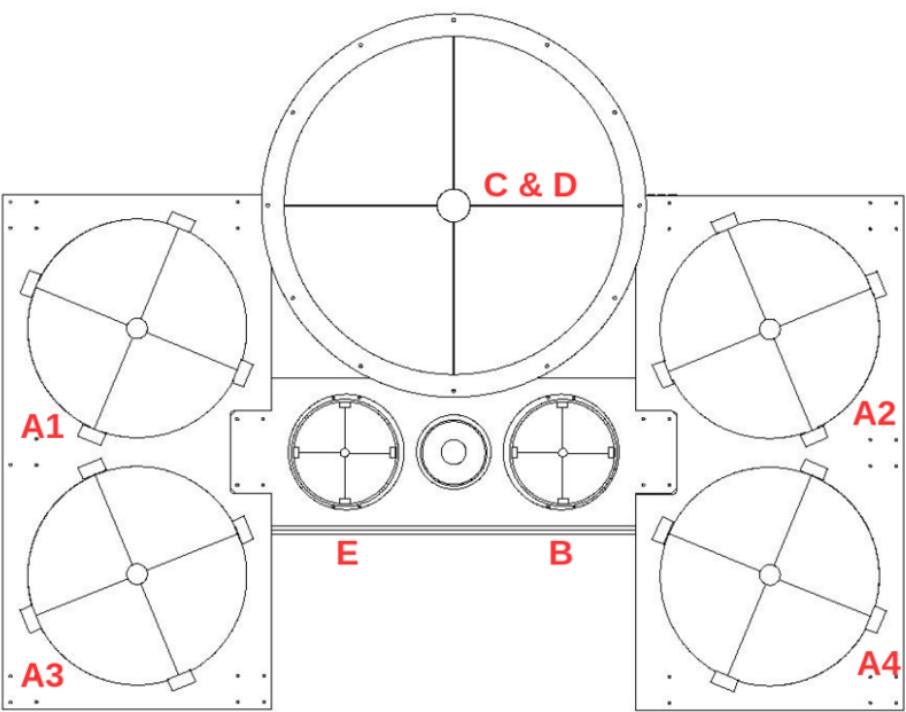

Figure 1: Mirrors A1, A2, A3, A4 (50 cm) are combined for the high gain Rayleigh channel. B (20 cm) is low gain Rayleigh channel. Mirror C&D (80 cm) is the Raman channel for water vapour and molecular nitrogen. E (20 cm) is the Mie channel. The beam expander for the transmitted laser source is between mirrors E and B.

## 2.2  DIAL Ozone System (LiO$_3$S)

The OHP Differential Absorption Lidar (DIAL), also referred to as Lidar Ozone Stratosphère (LiO$_3$S),
uses two lasers to make a measurement of the vertical ozone profile using the differential absorption
by ozone at two different wavelengths. The first laser is an XeCl eximer laser used to produce a 308
nm laser source with a maximum power of 10 W. The beam is passed through a 3X beam expander
and has a final divergence of less than 0.1 mrad. The second laser is a tripled Nd:YAG which is used
to produce a 355 nm laser source with a maximum power of 2.5 W. The beam is passed through a
2.5X beam expander and has a final divergence of less than 0.2 mrad. Both transmitted beams have
a repetition rate of 50 Hz, and a 7 ns pulse width.

The receiver assembly consists of four 53 cm mirrors each having a focal length of 1500 mm,
a field of view of 0.67 mrad, and an average parallax of 3100 mm. Each of these four telescopes
are focused onto an actuator-mounted 1 mm diameter fibre optic. The outgoing signals are bundled
before being passed through a mechanical signal chopper to block low altitude returns below 8 km
which would saturate the photon counting electronics. The combined signal is split using a Horiba
Jobin Yvon holographic grating with 3600 grooves/mm and a dispersion of 0.3 mm/nm. The light
from the grating is projected directly onto the photomultipliers for a high (92%) and low gain (8%)
Rayleigh channel at 308 nm, a high gain (92%) and low gain (8%) Rayleigh channel at 355 nm, and
two Raman channels at 331.8 nm and 386.7 nm for molecular nitrogen. The spectral resolution of
the light incident on the photo cathode is on the order of 1 nm. Figure 2 shows a schematic of the
OHP DIAL system.

All channels are sampled using a Licel digital transient recorder with a record time of 0.25 $\mu$s
which corresponds to a vertical resolution of 75 m. Further details can be found in (Godin-Beekmann
et al., 2003).

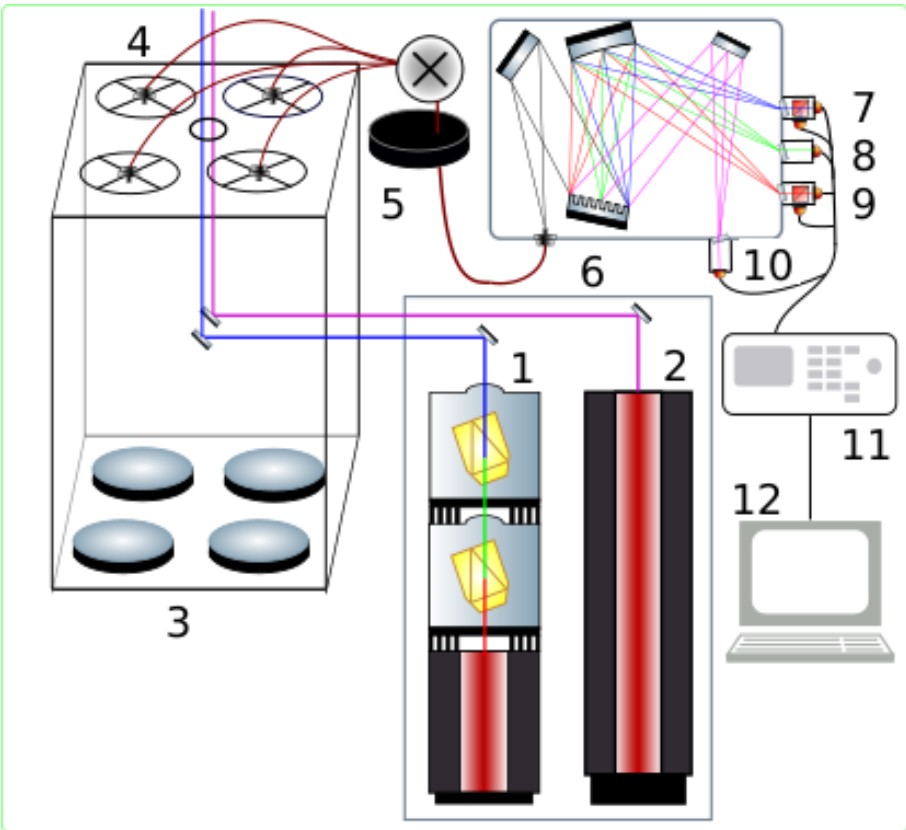

Figure 2: LiO$_3$S DIAL system. **1** 355 nm laser source, **2** 308 nm laser source, **3** four 530 mm mirrors, **4** four actuator mounted fibre optic cables, **5** mechanical chopper, **6** Horiba Jobin Yvon holographic grating, **7** 308 nm high and low gain photomultipliers, **8** 331.8 nm photomultiplier, **9** 355 nm high and low gain photomultipliers, **10** 386.7 nm photomultiplier, **11** Licel transient signal recorder, **12** Signal processing and analysis computer.

## 3   Methods

In this section we will set forth rigorous and well defined procedures for the retrieval of lidar temperatures in the middle atmosphere which will minimize the uncertainties at the upper limit of the lidar altitude range.

### 3.1   Rayleigh Lidar Equation

To calculate absolute temperature profiles from relative density profiles we exploit the gradient of the measured profile of back-scattered photons collected by the receiver. From classical lidar theory (Hauchecorne and Chanin, 1980), we know that the number of photons received is a simple product of transmitted laser power, atmospheric transmission, telescope geometry, and receiver efficiencies. This quantity can be expressed numerically in Eq. (1):

$$N(z) = \xi_{sys} \cdot \tau_{emitted}(z,\lambda) \cdot \tau_{return}(z,\lambda) \cdot O(z) \cdot P_{laser} \cdot \frac{\lambda_{laser}}{h \cdot c} \cdot \sigma_{cross} \cdot n(z) \cdot \frac{A}{4\pi z^2} \cdot \Delta t \cdot \Delta z + B$$

(1)

$N$ is the count rate of returned photons per time integration per altitude bin

$z$ is the altitude above the detector

$\xi_{sys}$ is the system specific receiver efficiency

$\tau_{emitted}(z,\lambda)$ is the transmittance of the photons through the atmosphere

$\tau_{return}(z,\lambda)$ is the return transmittance of the photons through the atmosphere

$O(z)$ is the overlap function of the receiver field of view

$P_{laser}$ is the laser power at a given wavelength

$\sigma_{cross}$ is the backscattering cross section of the target molecule

$n(z)$ is the number density of scatterers in the atmosphere

$\frac{A}{4\pi z^2}$ is the effective area of the primary telescope

$\Delta t$ is the temporal integration for data collection

$\Delta z$ is the spatial range over which photons in a bin are integrated

$B$ is the background count rate.

There are four simple assumptions we make when Eq. (1) is used. First, we assume that each photon we count only scatters once. While this is almost certainly not the case, we can say that it is approximately true. Visual wavelength photons have a very low probability of scattering in the atmosphere and with a multiple-scatter process we must square that very small probability. Of these multiply scattered photons, only those with a scatter angle towards the lidar receiver assembly will

be seen, with the vast majority scattering outside of the field of view. Further, the tenuous nature of the UMLT means that the small probability of detecting a photon which has scattered more than once becomes exponentially negligible with increasing altitude.

Second, we assume that the atmospheric density is directly proportional to the number of returned photons incident on the receiver assembly. In the case of high signal returns from the lower

atmosphere, when the number of returned photons can saturate the photon counting electronics, the measured photon count rate will diverge from the received photon count rate. Multiple detection channels, at different sensitivities, are used to compensate for this effect. In this work we are primarily concerned with the UMLT, a region where lidars operate at very low count rates, so for the purposes of this work we can safely make this assumption. A correction for saturation in the lower

stratosphere is described in Sect. 3.5.1.

Third, we assume that the atmosphere is in local hydrostatic equilibrium as well as local thermodynamic equilibrium (LTE) and obeys the ideal gas law. This assumption is potentially problematic at high altitudes where non-LTE processes can affect gravity wave dynamics and temperature profiles (Apruzese et al., 1984). However, given that a single lidar profile is acquired every 2.8 minutes

and a nightly average temperature is generated every 4 hours, we can have some confidence in this assumption.

Fourth, we assume that the atmosphere at mid-latitudes is generally free of aerosols above 30 km when there are no active volcanic or fire events (Hauchecorne and Chanin, 1980). During less severe background aerosol conditions (aerosol scattering ratio < 1.02), (Gross et al., 1997) suggests lidar temperature cold biases due to Mie scattering are less than 0.5 K at 20 km.

In the UMLT the signal to noise ratio and the model derived a priori assumptions for pressure and density are the main sources of error for the lidar temperature retrieval method. This paper lays out a rigorous method for reducing the noise in this region of the lidar signal with the goal of producing more robust mesospheric temperatures.

## 3.2 The Raw Counts Lidar Signal

When backscattered photons are incident on the lidar receiver they are integrated for a set period of time in the counting electronics. This ensures that the recorded signals are based on a similar number of transmitted photons. In the case of LTA a photon count profile, as a function of arrival time, is generated for every 5000 laser shots. Similarly for $LiO_3S$ a photon counts profile is produced for every 8000 laser shots. These measurements can be further integrated for the entire night to increase the signal to noise ratio at the upper limit of the measurement range. We use the speed of light to convert our profiles of photon count rate per second as a function of arrival time at the detector to total photon count rate per second as a function of altitude.

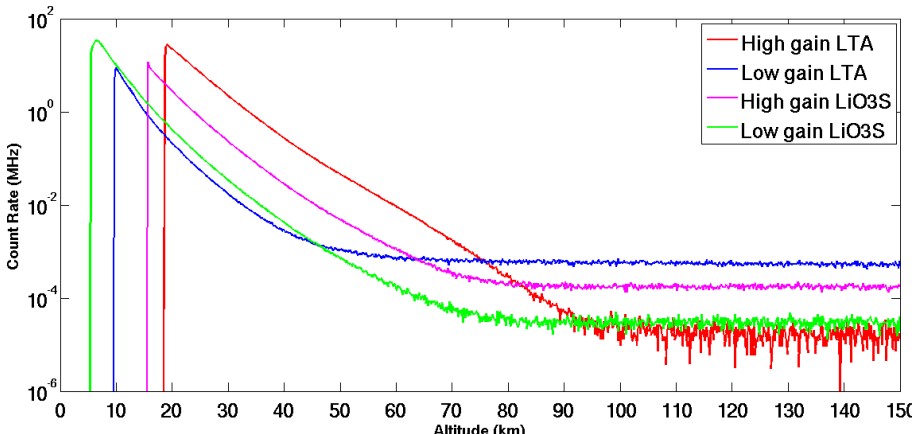

Figure 3: Nightly integrated profiles for high and low gain Rayleigh signals for LTA and $LiO_3S$. The background for LTA extends to 246.23 km and for $LiO_3S$ extends to 154.13 km. A single lidar profile for both LTA and $LiO_3S$ has a temporal resolution of roughly 2 minutes and 45 seconds and a vertical resolution of 75 m.

Figure 3 shows four nightly integrated OHP lidar count rate profiles as a function of altitude. Both lidar systems employ a high gain and a low gain channel to extend the measurements over a greater altitude range. The lower altitudes (corresponding to the fastest signal return times) of each

channel are either blocked by a mechanical chopper or electronically blanked. This is done to avoid saturation of the receiver assembly from very large signals in the lower atmosphere. Additionally, each channel has a set of optics designed to minimize the noise, with greater care being given to the high gain channels. These optics are fully described in the instruments Sect. 2.

### 3.3 Identifying Outliers, Signal Spikes, Signal Induced Noise, and Transient Electronic Interference

When retrieving lidar temperature profiles in the UMLT it is necessary to take extra precautions to carefully remove outliers, spikes, and electronic contamination from each profile in both the background region and the signal regions. Any contamination of the signal in the background region will be of the same order of magnitude as the true signal and thus, have a disproportionate effect on the temperature. An overestimation of the background due to localized signal contaminations will result in the removal of true photons, a lower estimated density, and by the ideal gas law, a higher temperature. The shape of the temperature profile itself will be distorted if there is a non-constant background. If it is not possible to fully correct the issue it is highly recommended to exclude the entire profile from the nightly analysis.

#### 3.3.1 Spikes

Spikes in fast integration photon counting data are not always easy to spot but can be defined as anomalously large, isolated, signal rates which occur in only one altitude bin without affecting adjacent data. If not properly identified and extracted from the data they can contribute to false temperature features and inaccurate background estimations. The spikes can have many potential origins (thermal or electronic imperfection in the photomultiplier, small charges in the Licel digital recorder, interaction of the photocathode substrate with a cosmic ray, or dozens of different kinds of electronic 'cross-talk' between all the instruments at the observatory station) and are therefore impossible, in practical terms, to completely prevent in the lidar data set, and completely impossible to prevent in measurements which have already been made. Therefore, it is necessary to address this problem using software during the analysis. It is particularly challenging to separate small amplitude spikes when the signal to noise ratio approaches 1. It is therefore necessary to establish a consistent criterion to determine which data points belong to the the population of real lidar returns and which points are likely contamination spikes. We have chosen to employ a straight forward Tukey Quartile test (Tukey, 1949) on the difference between consecutively binned lidar returns as this statistic is relatively insensitive to signal drift during the course of the night. The quartile technique is equally useful in both regions of high signal returns as well as the background regions and shows stability and consistency in identifying outliers. Figure 4 is a plot of photon count rate as a function of binned arrival time and shows an example of several photon count acquisitions plotted as a stack plot with

the black line representing the $2\sigma$ limit on the population of lidar returns. Data points above the black line are considered as signal contamination and are removed from the analysis.

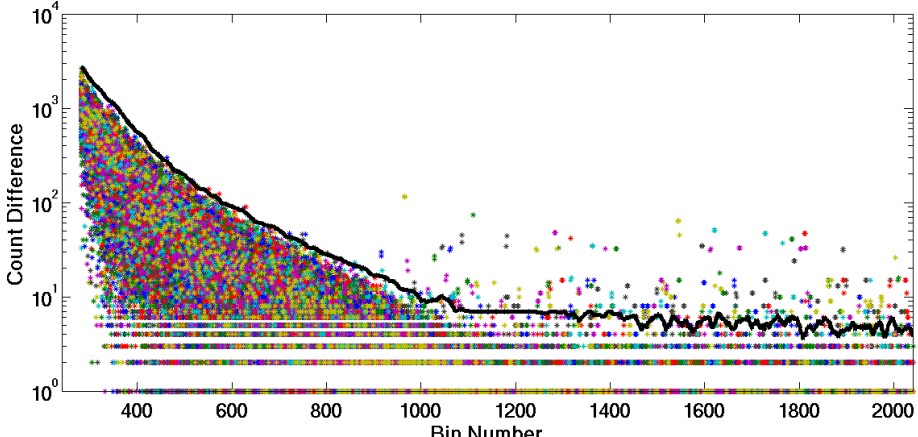

Figure 4: Tukey Quartile spike identification based on the signal difference between consecutive lidar time bins for short integration lidar returns. An entire night of lidar profiles is over-plotted in the stack plot. The black line is the 2 sigma limit and points above this line are removed.

### 3.3.2 Transient Electronic Signals

Transient Electronic Signals (TES) are short lived bursts in the lidar acquisition chain and may be internal to the system or related to nearby electronic interference. Possible sources for these transients include photomultiplier ringing from signal saturation, voltage fluctuations in the power supply, ambient RF signals, and ground loops between lidar electronics and Ethernet switches with metal sheathed cables. While these events are rare they can drastically alter the background and resulting temperature profile by inducing wavelike structures into the data.

Unlike simple spikes these features have an amplitude, a duration, and an effect on the counting rate in bins subsequent to the TES burst. In the example shown in Fig. 5 is a surface plot of counts differences between consecutive altitude bins for the first 100 altitude bins of lidar data. Each bin is 0.1 μs wide. This plot shows profiles for a night of lidar data with each profile accounting for roughly 1.6 minutes of lidar data. We can see that the 22nd and 46th profiles are contaminated by a TES with a duration of about 0.5 $\mu$s. These signals cannot be detected using the Tukey Quartile test as the time derivative of the photon return signal may not be sufficiently far from the nightly population median. However, a 2-D kurtosis test will consistently detect this type of signal contamination as a TES will induce a large skew in the photon count rate population distribution. The kurtosis test is done in the time dimension as well as with altitude to exclude false positives in the photon count rate skew which may be due to clouds or aerosols. Figure 5 (bottom) shows a plot of the kurtosis in the population of photon counts in each lidar profile and the red line shows the $2\sigma$ estimation of total lidar profile skew. Isolated profiles with a total kurtosis above this limit are excluded.

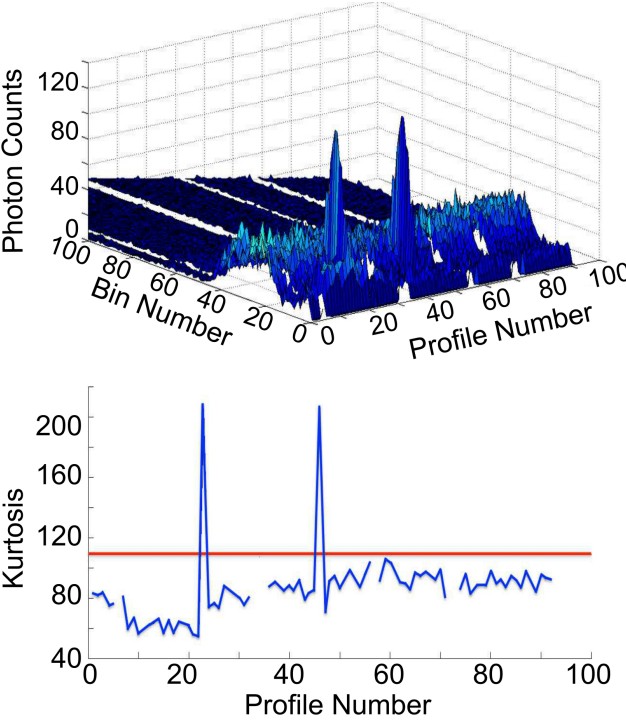

Figure 5: Upper panel is a surface plot of lidar returns as a function of time bin and profile number. For clarity, only the first 100 bins are shown in this plot. The test is carried out using all bins of each profile. Two instances of TES can be seen as anomalous peaks in the photon count rate. Lower panel is a summation of the fourth statistical moment (kurtosis/skew) for each scan. The red line indicates a $2\sigma$ limit on the skew of the population. Points above the limit are excluded.

### 3.3.3 Bad Profiles

After the removal of lidar profiles which suffer from clear signal contamination, there may still be profiles which ought not be included in a lidar temperature analysis because they are outliers of poor quality compared to other profiles within the same night. Conceptually, 'bad profiles' are lidar profiles with a high background and/or a low signal strength as compared to profiles measured shortly before or after the profile in question. These profiles need to be positively identified as not belonging to the general population of nightly lidar profiles. Quantitatively, identifying a 'bad profile' is a challenge as both the background and the signal can change abruptly over the night as the laser power drops or sky conditions change (see Fig. 6 for an example). In the top panel of the figure we see the evolution of the background for a night of lidar data. We might suggest that profiles 1 through 23 and profiles 36 through 46 might belong to one population and the rest (excluding profile 69) belong to a second population. However, when we look at the panel representing the signal, it is equally reasonable to, instead, interpret the plot as containing four groups. Each of these groups has similar signals which match fairly well with the changes in the backgrounds shown in the panels

above (profiles 1-23, profiles 24-35, profiles 36 - 48 and profiles 49 - 92) . However, whether these four groups of signals should be treated in analysis as two, three, or four distinct populations is open to interpretation. Therefore, we seek an objective programmatic solution for identifying bad profiles.

We now show two approaches for attempting to address the issue of changing signal quality. In Fig. 6 the green margin is an attempt to identify 'bad profiles' based on a moving average approach however, this method cannot accommodate quick transitions in signal strength and results in false positives when signal quality changes abruptly.

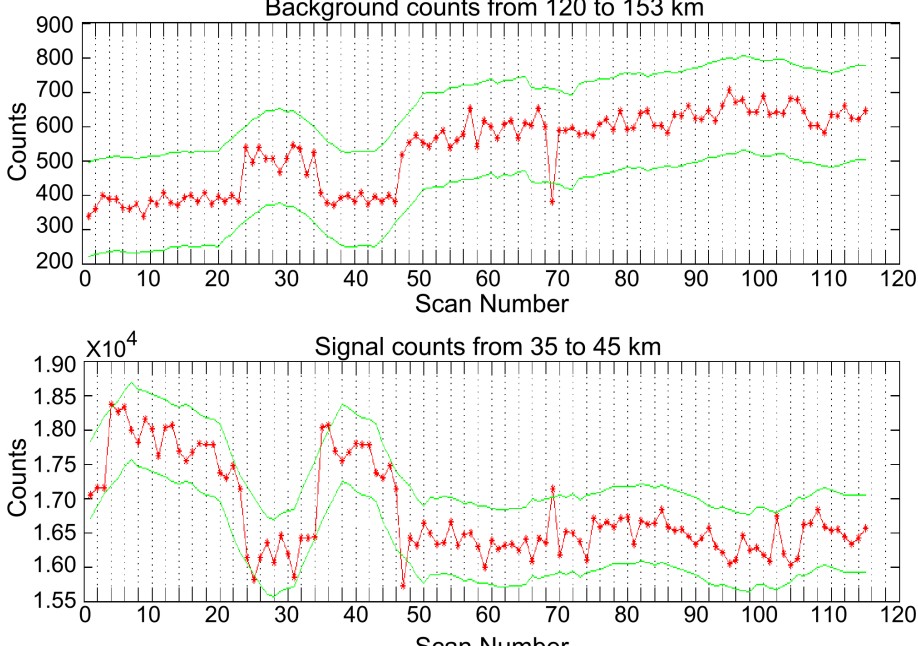

Figure 6: Example of lidar signal and noise during a night of measurements. Top panel shows the total background counts summed from 120 km to 153 km and the bottom panel shows the total signal summed between 35 km and 40 km. Green bounds are calculated based on a smoothed $2\sigma$ error estimation of the summed photon counts.

The simple reality of ground based observation means that lidar signals clearly detect changes
in the viewing conditions such as moonrise, thin cirrus clouds, optically thick clouds, changing light pollution, as well as changes in signal quality. Systematically identifying outlier signals is further complicated as there can be multiple signal to noise population medians during the course of the night. To properly characterize the non-Gaussian distribution of profiles and determine which should be excluded we require a non-parametric statistic. We use a one sided non-parametric Mann-
Whitney-Wilcoxon rank-sum test (Mann and Whitney, 1947) to identify lidar profiles which do not belong to the nightly population or subpopulations of lidar profiles.

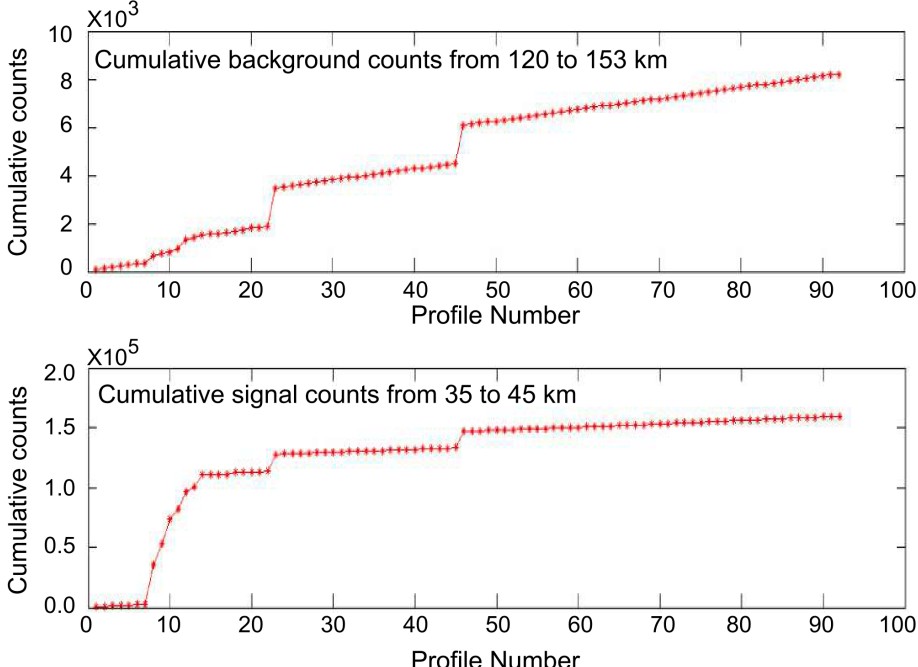

Figure 7: Rank sum plots for a night of lidar data. Top panel is the cumulative background count and the bottom panel is the cumulative signal count. The signal to noise ratio of the rank summed photon counts in each profile is evaluated using a Mann-Whitney-Wilcoxon rank-sum test to determine if an individual lidar profile belongs to the nightly population of lidar profiles.

Figure 7 shows the ranked sum of the background (noise) and signal counts for a night of lidar data. We do not exclude the profiles which fail the test for having high quality. The benefit of using this metric is that it allows us to have a standardized definition of a 'bad profile' which takes into account the nightly median without the assumption that the quality of lidar profiles is normally distributed. In this example the first 13 profiles fail the rank-sum test and are discarded.

### 3.3.4 Good Profiles

Given that our objective is to calculate accurate temperature profiles at the highest possible altitudes we must quality test each profile that we choose to include in the nightly average. It is possible to include partial profiles but that is not done in this work. The conceptual difference between a 'bad profile' and a 'good profile' is that bad profiles are positively identified as outliers to the general population whereas good profiles represent the portion of the population of profiles which contribute more information than noise to the nightly average at a given altitude. Consider that a poor quality lidar profile which has a signal to noise ratio of 1 at 70 km contributes more information from the signal than from the (background + noise) at 60 km, but less information from the signal than from

the (background + noise) at 80 km. Thus, we need a flexible metric to determine signal quality over a diagnostic altitude which reflects the general signal quality of the night.

Quantitatively, we express this with a signal, $S$, to noise, $N$, inequality in Eq. (2). The background (noise) of an individual profile, $N_i$, is expressed as the summation of photon counts in bins which fall between 120 km and 155 km and the nightly background, $N_{sum}$ is the summation of all $N_i$ for the night. To determine a metric for the nightly average lidar signal, $S_{sum}$, we first calculate a quick density profile and determine the lowest altitude where the signal to noise ratio equals 1. We chose a cutoff value of SNR=1 because it is the least strict value we could use which ensures that we have more information than noise (or, specifically, more information than noise plus background counts), at the altitude within the density profile where we begin the downward temperature integration. Had we chosen a criterion which was less strict (SNR≪1), we would expect to see more statistical variability in the top altitudes of the temperature retrieval as a result of starting the temperature integration in a region which contains more noise than signal. Conversely, choosing a criterion which is too strict (SNR≫1) limits the maximum altitude of the temperature retrieval as discussed in Sect. 3.6.1. The SNR = 1 point forms the upper bound of the altitude range from which we derive the representative signal for the profile. The lower bound of this representative signal range is defined to be one density scale height (∼8 km) below the upper bound. The lidar range bins which correspond to this altitude range are then summed to yield $S_{sum}$. A similar calculation, using the same range bins as in the nightly average calculation, is done to determine the signal of a single profile, $S_i$. If a profile fails the inequality test then it is not included in further nightly analysis.

$$\sqrt{\frac{S_{sum} + N_{sum}}{S_{sum}}} < \sqrt{\frac{(S_{sum} - S_i) + (N_{sum} - N_i)}{S_{sum} - S_i}} \qquad (2)$$

### 3.4 Noise Reduction

Statistical uncertainty in photon counting can be described by a Poisson distribution based on the square root of the number of photons received. Systematic uncertainties in the photon counts are introduced by ambient background light (light pollution, moonlight etc.), thermal excitation in the photomultipliers (so-called dark current), and signal induced noise. The first two sources of error are minimized by using narrow filters in the optical receiver chain and by cooling the photomultipliers. The signal induced noise can be very difficult to correct experimentally and is usually estimated in data processing. This type of noise can occur if the photomultipliers have become saturated at any point in the signal acquisition process and often manifest as non-linear artifacts superimposed upon the true photon count profile.

Figure 8 shows the reduction in the background noise due to recent hardware improvements. The first drop corresponds improvements made to the photomultiplier cooling system which reduces the number of thermally excited electrons detected at the photo cathode of the photomultiplier in the absence of signal from the sky. The second drop in background counts results from replacing the Hamamatsu R7600U-20 multi-alkali photomultiplier with the improved Hamamatsu R9880U-

110 photomultiplier having a super bi-alkali photo-cathode. The third and final drop in background counts is a result of replacing a 532 nm optical filter which has a width of 1 nm with a newer filter having a bandwidth of 0.3 nm. These experimental modifications result in a 100 fold decrease in the background noise and allows us greater confidence in our UMLT temperature retrievals. The regular monthly variations in the signal which become apparent at lower noise levels are due to the phase of the moon.

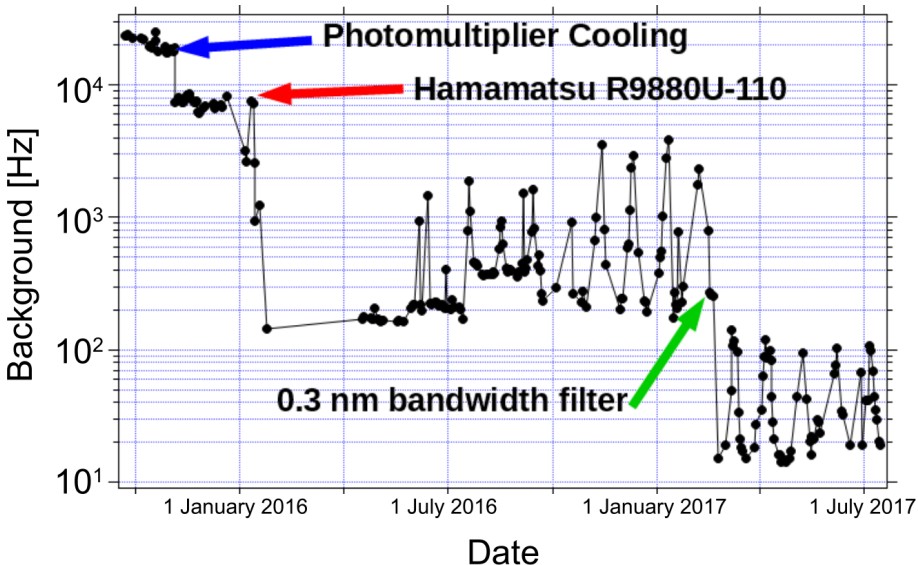

Figure 8: This figure shows the improvements in the background count rate due to photomultiplier cooling, new photomultipliers, and new optical filters. Note the logarithmic y-axis and the total reduction of background counts by more than 2 orders of magnitude.

### 3.5  Corrections Applied Before Temperature Calculation

In the previous subsection we detailed the process for removing bad data points and profiles from our nightly lidar measurement. In this subsection we will detail several corrections to our remaining photon count profiles which correct for signal saturation, atmospheric transmission, and background estimation.

#### 3.5.1  Deadtime Correction

The OHP lidars measure photons using photomultipliers and a digitizing signal counter. This system is highly efficient at detecting low signals and is optimized for single photon returns in the UMLT. However, given that the returned lidar signal directly follows the exponential density of the atmosphere, the photomultipliers and counting systems are susceptible to missing photons at lower altitudes due to high count rates. To correct for this saturation effect we can estimate a correction coefficient, $\tau$, also referred to as a deadtime.

The background theory and derivation of Eq. (3) is well described by (Donovan et al., 1993), where $N_{received}$ is the number of photons incident on the PMT per measurement time interval and $N_{counted}$ is the number of photons per measurement time interval which are actually counted by the system. In general, $N_{counted} < N_{received}$ due to effects of the system deadtime. This deadtime correction can be calculated based on factory specification of the counting electronics, a theoretically derived

deadtime, or it can be measured directly using a low gain lidar channel. The OHP lidars measure the deadtime directly and correct for saturation in the high gain channels with information from the low gain channels. If the low gain channel is not available a theoretical correction of 7 ns is applied to pre-2013 data and 4 ns is applied to more recent data following the installation of a Licel digital recorder.

In order to measure the deadtime experimentally, we assume that the low gain channel, because it has low photon count rates, will always operate in the linear response regime and will never suffer from deadtime effects. Thus, it represents a value proportional to the 'true' rate for returned photons for each altitude. Once scaled by a constant (e.g. using MSIS or another model), we can use this count rate as $N_{received}$.

The high gain channel, conversely, measures higher photon count rates at every altitude than the low gain channel does. Similarly to the low gain channel, at the low end of its dynamic range, the high gain channel operates linearly, and therefore represents a value proportional to the 'true' rate for returned photons for each altitude. The constant of proportionality is different for low and high gain channels. At low count rates, the scaled counts measured by the high gain and low gain channels are

equal. As photon count rates move into the higher end of the high gain channel's dynamic range, deadtime begins to have an effect: The high gain channel will measure too few photons compared to the 'true' rate; the number of photons which are returned to the lidar. Therefore, we call the scaled high gain count rate $N_{uncorrected}$ in Eq. (3); it has not yet been dead time corrected. We will refer to the deadtime corrected scaled high gain count rate as $N_{dtc}$. Equation (3) is used several times.

First, we use data only from altitudes for which the low gain and high gain channels both have measurements (nominally 40 to 60 km). We iterate through various values of $\tau$, calculating a $N_{dtc}$ for each $N_{uncorrected}$ value. This is carried out until the difference between $N_{corrected}$ (from the high gain channel) and $N_{received}$ (from the low gain channel) is minimized. This determines the dead time of the system, $\tau$. Next, Eq. (3) is used again, using the measured nightly value for $\tau$, to

calculate $N_{dtc}$ for all $N_{uncorrected}$ high gain channel measurements. This allows us to correct the high gain measurements for the entire profile.

$$N_{dtc} = N_{uncorrected} * exp(\frac{\tau * N_{uncorrected}}{\Delta t}) \tag{3}$$

### 3.5.2 Atmospheric Transmission Correction

To correct for Rayleigh extinction we use MSIS-90 model (Picone et al., 2002) to generate a vertical

profile of ozone, molecular oxygen, oxygen radical, molecular nitrogen, and argon, and then apply

the correct Rayleigh cross-section to each species. This method is adapted from (Argall, 2007) and is important for accurate retrievals of density and neutral temperature in the UMLT. Correction for aerosols is not done in this work as we assume that the atmosphere is generally clean above 30 km (Hauchecorne and Chanin, 1980).

### 375    3.5.3   Defining the Background

Normally, we assume that the rate of counted photons per laser shot is constant in the background region during the signal acquisition time and can therefore be approximated by a simple Poisson distribution. We further assume that in this background region we are not measuring returned photons from the laser signal but instead are measuring ambient sky light. However, if there is non-linear
signal induced noise in the photon counting chain, the number of counted photons is not constant with time during the acquisition period of a single laser shot. When this occurs we cannot assume that the variation in the background is a strictly Poisson distribution around a constant expected value.

If left uncorrected, we risk overestimating the number of 'true' photons returned from the upper
atmosphere and the result is an artificially dense and cold UMLT. Erring on the side of caution we fit three backgrounds (constant, linear, and quadratic) to each nightly summed profile, in a standard diagnostic region, and choose the function with the best Chi-squared goodness of fit as our estimate of signal induced noise. The best background function is subtracted from the raw photon counts profile. Shown in Fig. 9 is an example of a night where the low gain Rayleigh channel (blue) experienced
signal induced noise which was best approximated by a quadratic function; the high gain Rayleigh channel (red) had a background best estimated by a small negative linear function; and the nitrogen Raman channel (green) had no apparent signal induced noise and was fit with a constant background. The optimal solution for non-linear signal induced noise is to determine the contribution of both the signal and the noise using exponential fits however, we have found that method to be extremely sen-
sitive to the choice of background diagnostic region and was less stable than the simple quadratic approximation.

We have some confidence that the quadratic background correction to the low gain channel correctly approximates the moderate non-linear signal induced error because we can compare the corrected low gain channels to the high gain channel. In the overlap region we have two channels making
coincident measurements and we can safely assume that the response rate for the high gain channel is linear. Therefore, a correction for signal induced noise in the low gain channel which brings the resulting low gain count rates into the closest agreement with the high gain channel count rates at the same altitudes will be the optimal choice for the correction. In some cases, the quadratic correction for signal induced noise in the low gain channel yields better agreement than the constant or linear
corrections, in which case it is employed. The best individual choice (constant, linear, quadratic) is used for each profile. We believe these empirical corrections to be sufficient, because (a) the re-

sulting agreement with the high gain channel improves as compared to the uncorrected profile, (b) the resulting corrected low gain count profiles are generally equal to the high gain count profiles to within statistical uncertainty, and (c) for the few cases in which the empirical correction ultimately

fails, this will be apparent by the corrected signal retaining poor SNR values. The melding procedures of Section 3.6 weight the combined high and low gain Rayleigh channels according to SNR, and so in these cases, the poorly-corrected low gain contributions to the final melded counts profile will be negligible, and all information will be obtained from the high gain channel.

For the quadratic case, as soon as there is signal induced noise the profiles no longer represent

Poisson distributions as the count rate in each lidar bin is no longer fully independent of the count rates in the bins on either side of it. Therefore, precise calculations of the SNR would require the addition in quadrature of real noise (from sky background and signal photon counts) and contamination noise (from signal induced noise). Here, however, we make the assumption that the signal induced noise is able to be completely removed from the raw profiles with the subtraction of the

quadratic function. We therefore interpret the background subtracted profiles to obey approximately Poisson distributions, thereby approximating the total noise in the profile to the noise of only the real photons, which can be treated as uncorrelated. Our standard altitude range for background selection is 120 km to 155 km but this number is system and channel specific. To illustrate this point we compare the background regions of the high gain Rayleigh channel (red) and the nitrogen Raman

channel (green) in Fig. 9. The nitrogen Raman channel background could be calculated from 50 to 155 km or 120 to 155 km and yield the same result.

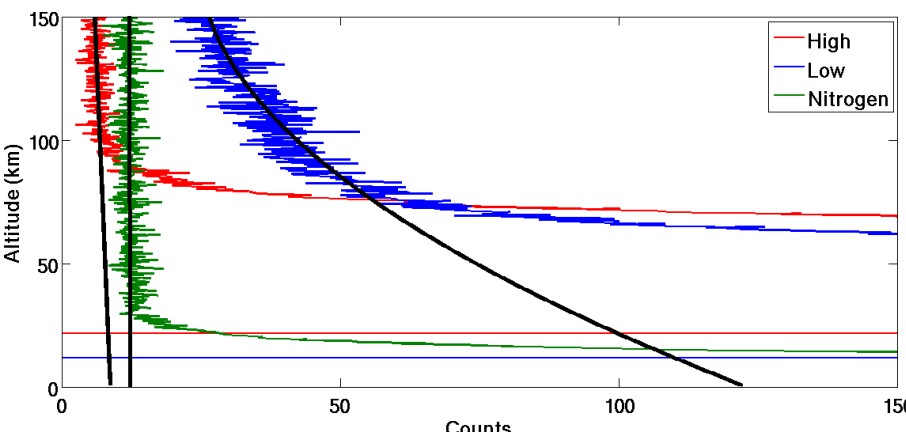

Figure 9: An example of non-linear signal induced noise in the low gain Rayleigh channel best estimated by a quadratic background. Also shown is the high gain Rayleigh channel (red) with a background best fit by a negative linear function and the nitrogen Raman channel (green) with no apparent signal induced noise and a constant background.

### 3.6 Temperature Inversion Equation

The standard NDACC algorithm for Rayleigh temperature retrieval is the Hauchecorne-Chanin (HC) method (Hauchecorne and Chanin, 1980) which makes a scalar normalisation of the photon-count profile to an in-situ density measurement or to a density calculated from a model like CIRA-72, SPARC-80, or MSIS-90. From a density gradient profile we calculate a pressure gradient profile Eq. (4) and using the ideal gas law, Eq. (5), we can arrive at an expression for pressure, Eq. (6). Here $P$ is pressure, $z$ is altitude above the lidar station, $\rho$ is density, $g$ is the latitude dependent acceleration due to gravity for an ellipsoid Earth given by the Somigliana formula, $R$ is the ideal gas constant, $T$ is the temperature, and $M$ is the molecular mass.

$$dP(z) = -\rho(z)g(z)dz \tag{4}$$

$$P(z) = \frac{R\rho(z)T(z)}{M} \tag{5}$$

$$\frac{dP(z)}{P(z)} = -\frac{Mg(z)}{RT(z)}dz = d(\log(P(z))) \tag{6}$$

The crux of the challenge for initializing the lidar equation lies in the non-linear nature of Eq. (6) which will necessitate the introduction of an a priori estimate of pressure at the top of the atmosphere followed by an iterative approach to retrieving the profile at lower attitudes. A full theoretical description of this problem was well laid out by (Khanna et al., 2012). In this work we have chosen to take our initial a priori seed pressure value, $P(z_1)$, from the MSIS-90 model. We now arrive at an iterative expression for the generation of the pressure profile as a function of altitude Eq. (7).

$$\frac{P(z_i) - \frac{\Delta z}{2}}{P(z_i) + \frac{\Delta z}{2}} = \exp \frac{Mg(z_i)}{RT(z_i)}\Delta z \tag{7}$$

Given our iteratively generated pressure profile we can do an inverse calculation to map our pressures to a set of temperatures using Eq. (8) and Eq. (9). This iteration starts at the top of the atmosphere, in a region of low signal to noise and thus of large relative uncertainty, and proceeds downwards in altitude and becomes exponentially less uncertain with each step as signal quality improves with increasing atmospheric pressure. As we iterate downward the influence of our choice of a priori pressure becomes less significant and the calculated temperature profile becomes entirely data driven.

$$X_i = \frac{\rho(z_i)g(z_i)\Delta z}{P(z_i) + \frac{\Delta z}{2}} \tag{8}$$

$$T(z_i) = \frac{Mg(z_i)}{R\log(1 + X_i)}\Delta z \tag{9}$$

In order to calculate a single temperature profile from 5 km to above 80 km we meld the photon counts from the high and low gain Rayleigh channels together with the counts from the $N_2$ Raman channel. The slope of the logarithm of each of the three photon counts profiles is compared to a

synthetic lidar counts profile generated based on the nightly average MSIS-90 density profile. The comparison gives us a first estimation of the linearity and alignment of the lidar data. We then select a clear linear region of each profile to use in calculating a MSIS derived scaling factor for each profile. This procedure allows the top of the nitrogen Raman profile to be melded to the bottom of the low gain Rayleigh profile and the top of the low gain Rayleigh profile to be melded to the bottom of the high gain Rayleigh profile. The melding calculation is conducted over a signal-to-noise defined altitude range and is a straightforward weighted average. The resulting melded density and pressure profiles are used to generate a single nightly average temperature profile like the one shown in Fig. 10. The use of MSIS-90 as a scalar density reference for the synthetic lidar profile does not affect the final lidar temperature profile which depends only on the relative density and not the absolute value. We follow similar procedures to those described by (Alpers et al., 2004).

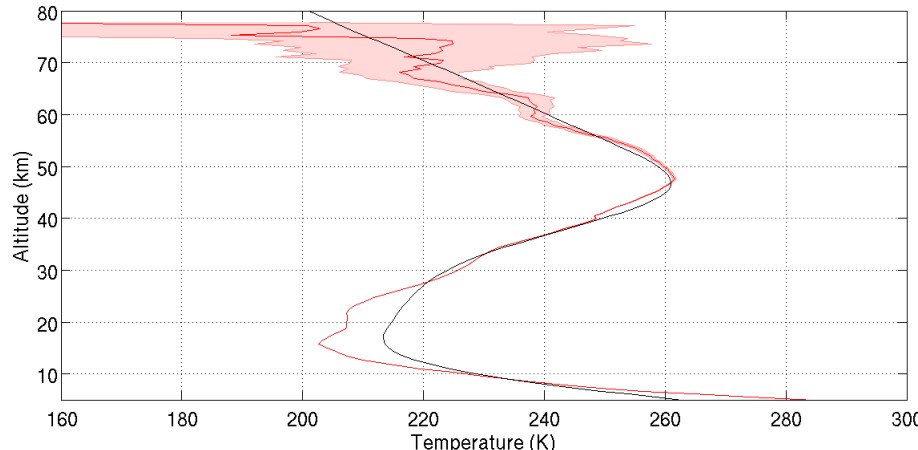

Figure 10: An example of a nightly average melded temperature profile from two Rayleigh channels and one Raman channel. The profile is calculated at 300 m vertical resolution from a single combined photon count profile and has a maximum relative error near 80 km of 30%. Black line is the MSIS-90 temperature profile which corresponds to the MSIS-90 pressure and density information we used as an a priori.

### 3.6.1  Where to start the inversion

As can be seen in Eq. (8) and Eq. (9) the calculation of lidar temperature requires an a priori guess of pressure at the top of the atmosphere and a relative density gradient. Given that the signal to noise in the UMLT can be very low, the choice of a priori as well as the uncertainties in the density gradient can have a very large effect on the temperature profile (Khanna et al., 2011). As a result, it is prudent to remove the top 15 km of the retrieval to minimize the contribution of the a priori (Leblanc et al., 1998b).

In our treatment the a priori pressure is selected at the altitude where the signal to noise ratio in a smoothed photon counts profile is 1. The resulting temperature profile is subsequently cut when the

relative error exceeds 30 percent. This treatment is not the optimal solution for the retrieval altitude as a fully Bayesian algorithm is required to properly characterize the influence of the a priori choice (Sica and Haefele, 2015). However, we believe that our signal to noise metric is sufficiently rigorous, and more importantly reproducible.

## 4 Net result of temperature algorithm modifications

The NDACC algorithm contains such corrections as deadtime, background, and transmission. The new algorithm improves upon the background correction and identification of bad profiles, and introduces corrections for: signal spikes, TES, identification of good profiles, and noise reduction, all which have not previously been addressed by the NDACC algorithm.

The LTA data is recorded and saved at 75 m resolution. The spike and TES corrections described in sect.3.3.1 and 3.3.2 are carried out at this resolution. Then the profiles are integrated to 300 m, at which point the remainder of the corrections in Section 3 are applied.

Temperature profiles using the new algorithm are calculated at 300 m resolution for LTA, and are plotted as the green line in Fig. 11. This is higher resolution than the standard NDACC temperature resolution, which is 1 km, smoothed to 2 km effective vertical resolution. The LTA NDACC-calculated temperatures (black line in Fig. 11) are plotted at 2 km effective resolution. By implementing the new algorithm, we have cooled the UMLT lidar temperature retrievals with respect to the standard NDACC temperature algorithm. The modifications cool the mesospheric retrievals by approximately 5 K near 85 km and 20 K by 90 km. There is no significant difference between the new and the NDACC algorithms for LTA below 70 km.

Temperature profiles calculated for $LiO_3S$ are all carried out using the NDACC algorithm at an effective vertical resolution of 2 km, and these are shown as the orange line in Fig. 11. Whereas the LTA NDACC algorithm results are warmer than the $LiO_3S$ NDACC algorithm results above 70 km, we now see that the LTA new algorithm results are cooled sufficiently that they more closely match the $LiO_3S$ measurements up to 78 km. Therefore the corrections for LTA proposed in the new algorithm represent a significant improvement over the LTA NDACC algorithm for altitudes above 70 km.

A comparison with temperature retrievals from the satellites MLS (red line in Fig. 11) and SABER (blue with shaded ensemble variance), and with the MSIS-90 model (magenta line in Fig. 11), also shows an improvement in the LTA temperatures retrieved using the new algorithm as compared to the LTA NDACC algorithm. By implementing the techniques described in the sections above we can account for nearly half of the temperature difference between the lidar and the satellites at 90 km. The character change in the difference functions above and below 84 km is in part due to the increasing contributions of the species specific Rayleigh backscattering correction and the corrections to the gravity vector. The remaining temperature difference between the improved lidar

temperatures (green) and the satellites and model may be in part due to distortions in the satellite a priori for the geopotential vector. This possibility is explored further in the companion paper, and all coincidence criteria for the satellite comparisons are available therein (Wing et al., 2018b).

It is important to note that additional complications exist when comparing temperatures derived from ground based lidars to temperatures derived from satellite data which have their own calibration concerns. We explore the issues of lidar-satellite comparison in part B of this paper (Wing et al., 2018b). A co-located ground-based resonance Doppler or Boltzmann lidar would provide a better comparison data set as resonance lidars have high signal to noise ratios above 75 km (Alpers et al., 2004).

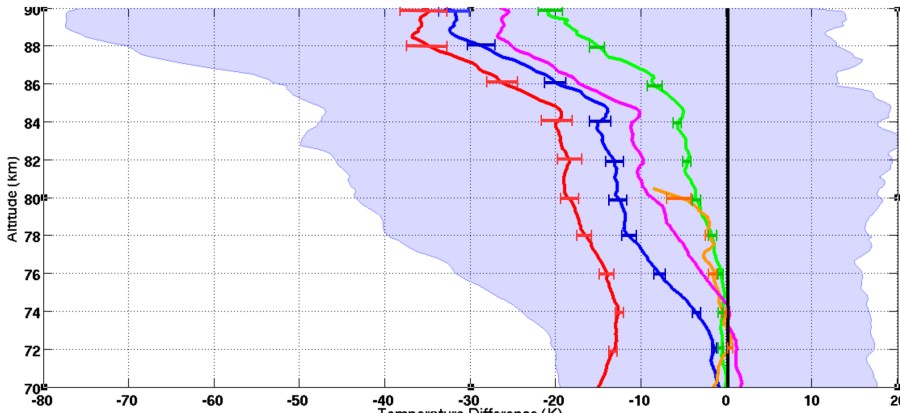

Figure 11: Ensemble temperature differences from NDACC standard LTA Rayleigh temperatures (black). MLS (red), SABER (blue with shaded ensemble variance), MSIS-90 (magenta), LiO$_3$S (orange), and LTA Rayleigh temperatures with corrections given in this work (green).

## 5 20 Year Comparison of OHP Lidar Temperatures

Conducting systematic inter-comparisons between independent lidar systems is essential for assuring data quality and is a requirement for NDACC certified instruments. Most comparisons are conducted on a campaign basis where two or more lidar systems are co-located and make coincident measurements. A good example of this type of work was the stratospheric lidar and Upper Atmospheric Research Satellite (UARS) validation campaign (Singh et al., 1996). The present study proposes a completely novel type of inter-lidar study on the long term stability of the Rayleigh lidar technique. The first step in our analysis is to compare the temperature profiles from the LTA and LiO$_3$S systems. LTA temperatures were calculated using the OHP NDACC temperature code and LiO$_3$S temperatures were calculated using a modified version of the same code. There are very few significant differences between these two codes. The most important difference involves the choice of parameters for melding the high and low gain channels for the two systems. Given the differences in the relative gain between the four lidar channels being considered, the melding of LiO$_3$S often

occurs at a lower altitude than LTA. The present study considers temperatures between 35 km and 75 km to ensure that we are well above any contamination from aerosols and below any significant initialization errors. From Fig. 11 we can see that there is no significant difference in the temperature outputs of these two algorithms (black baseline and orange) or with the improved algorithm (green) below 75 km.

We selected the data from 1993 to 2013 for the comparison as both instruments operated regularly and without significant design changes during this time. Since the lidars are co-located and are operated by the same technicians they often make measurements simultaneously. Figure 12 shows the average number of measurements per month made by the LTA and $LiO_3S$ which were included in this study as well as the average number of common measurements per month. We defined common 545 measurement times based on more than 80% temporal overlap, good quality profiles in both systems, and good internal alignment of both lidars. Of the 2482 nights of LTA data and 3194 nights of $LiO_3S$, 1496 nights met our criteria for coincidence.

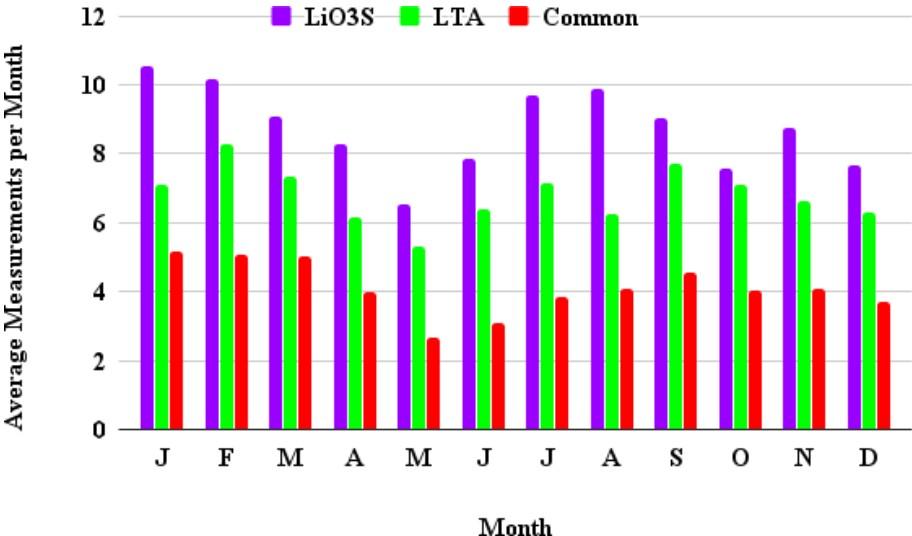

Figure 12: Average number of OHP lidar temperature measurements per month during the period of 1993-2013.

Figure 13 shows the nightly temperature differences between the two lidar systems. The 20 year data set contains 1496 coincident measurements lasting longer than four hours. Black vertical rect-550 angles indicate some of the time periods where the high or low gain channels were mis-aligned in one or the other lidar. Internal misalignments happen when one or more of the five mirrors in LTA or four mirrors in LiO3S are not properly aligned with the laser or the fibre optic is not centered on the focal point of the mirror. A few of these time periods can be associated with minor system modifications. Misaligned lidar signals were identified by comparing the slopes of the density pro-555 files in the high (generally above 50 km) and low (below ~50 km) gain channels of each system.

A simple chi-squared test was used to detect these nights and exclude them from the rest of the analysis. It is possible that the criteria described above for identifying periods of misalignment is not yet stringent enough. Therefore, one limitation of the OHP measurements in terms of accuracy, and depending on time scale, also precision, is the influence of periods of misalignment that have not been programmatically identified. An ideal solution would be to have an independent method of monitoring mirror alignment during atmospheric measurements (e.g. installation of a small sighting telescope to measure the alignment coupled with an automatic fiber optic alignment system). With the existing data set from OHP extending back two decades, we unfortunately cannot retrospectively address such a hardware goal, but there may be opportunities in future to look into the effects of choosing different criteria to identify periods of misalignment.

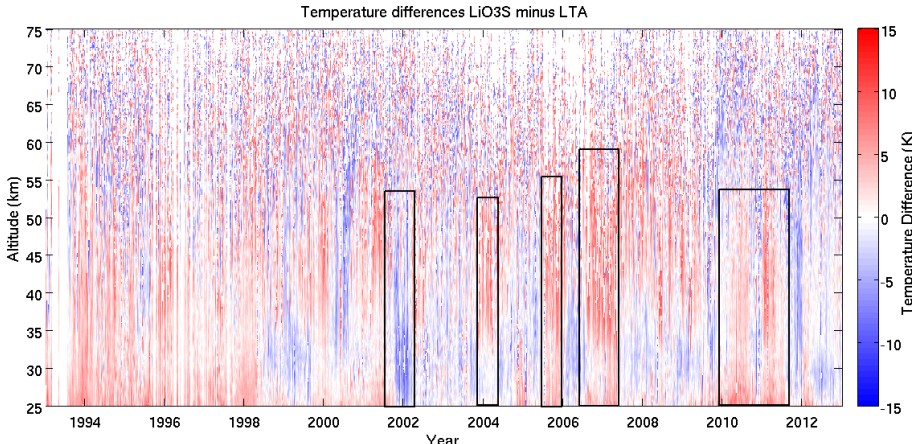

Figure 13: Temperature differences between LTA and LiO$_3$S OHP lidars for a 20 year period between 1993 and 2013. There are 1496 nights of comparison in this plot. Red indicates that LiO$_3$S was warmer than LTA and blue that it was colder. The black boxes highlight periods where the two lidars were out of alignment with respect to each other.

Figure 14 shows four curves depicting the average temperature differences as a function of altitude and year. The red curve is the average temperature difference between 65 km and 75 km with an average standard deviation of 6.6 K; the green curve is the average temperature difference between 55 km and 65 km with an average standard deviation of 4.5 K; the blue curve is the average temperature difference between 45 km and 55 km with an average standard deviation of 2.7 K; and the magenta curve is the average temperature difference between 35 km and 45 km with an average standard deviation of 1.6 K. A 30 day averaging window is applied to each of the four curves.

For reference, a typical LTA temperature profile with an effective vertical resolution of 2 km has an uncertainty due to statistical error of 0.2 K at 40 km; 0.4 K at 50 km; 0.6 K at 60 km; 0.7 K at 70 km; 1.8 K at 80 km; and 6 K at 90 km. For reference, a typical LiO3S temperature profile with an

effective vertical resolution of 2 km has an uncertainty due to statistical error of 0.3 K at 40 km; 0.5 K at 50 km; 1.0 K at 60 km; 2.7 K at 70 km; and 10 K at 80 km.

Examining the time evolution of the average temperature differences between LTA and LiO$_3$S at four altitude levels gives us confidence that both measurements are stable in both time and altitude. Using all data, including misaligned periods (example: winter 2006-2007 in Fig. 13 and Fig. 14) none of the lidar temperature differences are significant at the 2-sigma level, although certain periods do have temperature differences which are detectable at the 1-sigma level. This can be seen where the blue shaded region (2005 - 2008) and the magenta shaded region (in 2007) are entirely above the zero line. If the misaligned periods are disregarded, no temperature differences are significant, even at the 1-sigma level. Therefore, we conclude that the results from the lidars, when well-aligned, are stable in time, over the 20-year period studied.

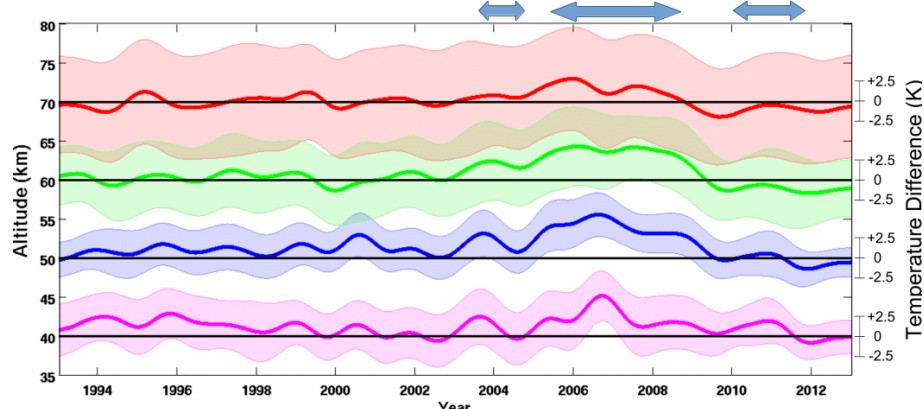

Figure 14: Average temperature differences between LTA and LiO$_3$S OHP lidars for a 20 year period between 1993 and 2013 at four altitude levels: 65-75 km (red), 55-65 km (green), 45-55 km (blue), and 35-45 km (magenta). Shaded uncertainties are shown at 1 sigma for clarity and the black lines are zero temperature difference displaced to 40, 50, 60 and 70 km. All measurements, including periods of lidar misalignment, are included in this plot. The apparent anomalies (blue arrows) occur only during times where the lidars were often misaligned, as indicated in Fig. 13.

After removing comparisons between mis-aligned instruments we can calculate the ensemble median difference between the two systems. The ensemble median difference in Fig. 15 shows very good agreement between the two co-located lidar instruments. The temperatures produced by LTA and LiO$_3$S are statistically equal above 45 km for the 20 year period between 1993 and 2013. There is a small –0.6 K systematic difference which reaches a maximum near 40 km. We believe this slight cold bias is due to small differences in the signal melding technique between the high and low gain channels in both systems. On a typical night, the LTA low gain channel starts to significantly contribute to the combined signal near 50 km. If the photon count rate in the low gain channel is too large at these altitudes (due to residual noise contributions or from a slight misalignment with the

high channel) the counts will be artificially higher than expected, resulting in a lower temperature. The converse holds true when the low gain channel is misaligned in the opposite sense, resulting in a slight warming due to underestimation of the counts.

The effect of these small temperature perturbations is so small that they can't be seen in single nightly temperature comparisons and were not detected before this study. It is important to note that the $2\sigma$ distribution about our ensemble at 40 km has a magnitude of approximately 0.45 K while the statistical error for a single night of lidar measurements near 40 km at 300 m vertical resolution can be on the order of 2 K. Detecting and resolving this small disagreement will be extremely challenging and will not be accomplished in this work.

Given that the primary interest of this work is the upper middle atmosphere (nominally above 50 km), we will focus on the upper portions of Fig. 15 where the two lidars are in statistically perfect agreement. To our knowledge, this is the first ever long term study of the temperatures produced by co-located temperature lidars operating at 532 nm and 355 nm. The excellent agreement between these two independent measurements gives us confidence that A) there is no vertical misalignment between the lidars, B) there are no unaccounted for optical transmission effects which influence our temperatures, C) the lidar measurements are reasonable and reproducible, D) we can now proceed with some confidence that our ground based lidar measurements can be useful as a calibration source for the space based satellite measurements.

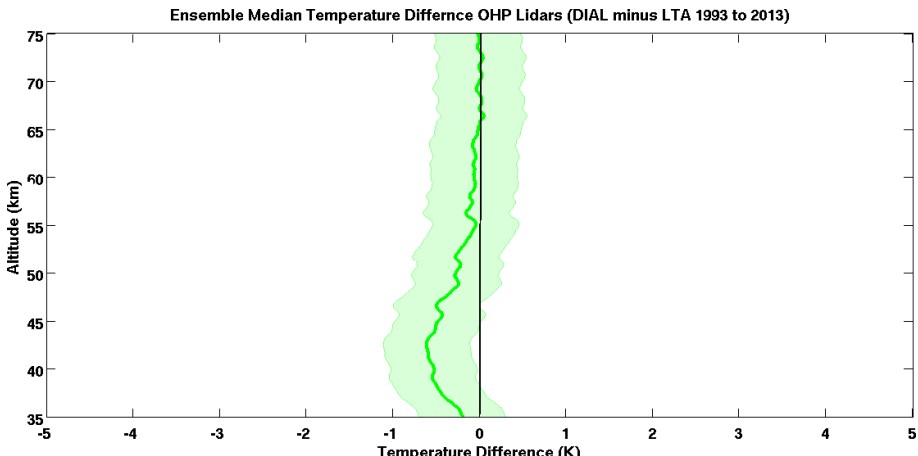

Figure 15: Ensemble of median temperature differences between LTA and LiO$_3$S based on temperature measurements between 1993 and 2013. Shaded error is the two sigma distribution about the ensemble.

## 6   Summary and Discussion

### 6.1   Changes to Lidar Temperature Algorithm

In this work we have attempted to minimize the systematic temperature bias at the top of the lidar temperature retrieval which has been noted previously by several studies cited in the introduction. We have done this by clearly and carefully outlining a rigorous, and complete algorithm for the calculation of lidar temperatures in the UMLT. We have presented techniques for the detection of
signal contamination, the selection of the best data for inclusion in the calculation, criteria for where to initialized the inversion when assuming an a priori pressure at the top of the atmosphere, and have demonstrated the benefit of photomultiplier cooling and narrow band pass filters to reduce lidar backgrounds.

After applying our techniques we have seen a systematic cooling of the high altitude lidar tem-
peratures which brings them into better agreement with the temperatures measured by both MLS and SABER (Fig. 11). It is also important to note the large variance associated with these ensemble differences can partially be attributed to the lack of control exerted on the error contribution from the choice of a priori initial pressure for lidar data and a priori contribution and non-LTE effects for satellite data. Part of the difference may also be due to altitude offsets and coarse vertical resolution.
Having applied these new data filtering techniques we have produced an improved lidar temperature data set which is exploited in the companion paper (Wing et al., 2018b) in an effort to validate satellite temperatures.

### 6.2   OHP Lidar 20 Year Comparison

We have conducted the first ever decadal temperature inter comparison between a co-located 532
635   nm Rayleigh lidar and an ozone DIAL system calculating temperatures from a 355 nm line. We have shown that:

**1)** Rayleigh lidar temperatures calculated from ozone DIAL non-absorbing 355 nm line are statistically equal to temperatures from a traditional 532 nm Rayleigh temperature lidar over a large altitude range. This finding is of particular interest for the NDACC lidar temperature database as
temperatures from ozone lidars may also be available for validation and inclusion.

**2)** Further theoretical work must be done on algorithms for melding data from high and low gain photon counting channels. The current techniques produce statistically identical nightly temperature profiles however, a -0.6 K bias near 40 km becomes apparent when multiple years of data are compared. It is doubtful that current data processing techniques can be easily adapted to address
this problem. However, an iterative, cost minimizing, Bayesian approach such as the one proposed by (Sica and Haefele, 2015) would be able to produce a single melded temperature profile with the accompanying averaging kernels and an estimate of the error due to the photon count melding. As a lidar development note, Fig. 13 demonstrates the need move towards the use of automated nightly

alignment of lidar system optics. Manual alignment by operators appears to lack consistency over
the time frame of multiple decades.

**3)** The two independent lidars show no evidence of significant instrument drift over a 20 year
period. This means that ground based lidars are the ideal choice of instrument for detecting small
calibration drifts in satellite remote measurements over long time scales. We rely on this finding to
justify the use of lidars as a reference data set for satellite validation in the companion paper Wing
et al. (2018b).

**4)** There is no evidence of a relative vertical offset between the two independently calibrated
lidar systems which would be seen as an 'S' shaped temperature bias in Fig. 15 due to the sign
change in temperature vertical gradient at the stratopause (Leblanc et al., 1998a). Based on personal
communication, recent July-August 2017 and March 2018 NDACC Ozone validation campaign at
OHP (LAVANDE) revealed no vertical shifts between either OHP lidar and the NASA STROZ
mobile validation lidar (McGee et al., 1995).

*Acknowledgements.* The data used in this paper were obtained as part of the Network for the Detection of Atmospheric Composition Change (NDACC) and are publicly available (see http://www.ndacc.org, http://cdsespri.ipsl.fr/NDACC) as well as from the SABER (see ftp://saber.gats-inc.com) and MLS (see https://mls.jpl.nasa.gov) data centres for public access. This work is supported by the Atmospheric dynamics Research InfraStructure Project (ARISE 2) which is funded by the European Union's Horizon 2020 research and innovation programme under grant agreement No. 653980. French NDACC activities are supported by Institut National des Sciences de l'Univers/Centre National de la Recherche Scientifique (INSU/CNRS), Université de Versailles Saint-Quentin-en-Yvelines (UVSQ), and Centre National d'Études Spatiales (CNES). The authors would also like to thank the C. Wing for graphics support, and the technicians at La Station Géophysique Gérard Mégie at OHP.

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
