# Peer review of "Lidar temperature series in the middle atmosphere as a reference data set. Part A: Improved retrievals and a 20-year cross validation of two co-located French lidars"

_Atmospheric Measurement Techniques, 2018_

## Referee Comment (RC1) · Anonymous Referee #1 · 1 Jun 2018

General Comments:

This manuscript presents one of the first thorough studies of issues associated with lidar temperature data retrieval using Rayleigh integration technique. Authors have done nice work to come up with systematic methods to improve data screening using various statistical and signal processing techniques. Authors have also spent tremendous efforts in addressing various issues in the data retrieval procedures, such as the seeding temperature/pressure, atmospheric transmission correction, and the determination of starting altitude, etc. Furthermore, authors compared 20 years of Rayleigh temper-

ature data between two French lidar systems, setting a good foundation for Rayleigh lidar temperatures to be good references for satellite and other missions. As many Rayleigh lidars are being deployed all over the world to make atmospheric measurements, this manuscript just came in time to make people to be aware of potential issues and help people improve the quality of retrieved temperature results.

However, this is a long paper printed with tiny fonts, so it is understandable that there are quite some technical issues along with clarifications needed. I recommend acceptance of the paper for publication after technical corrections.

Technical and Specific Comments (going by page number):

1) Page 6: Lidar equation (1) has dimension mismatch on the left and right hand sides. The first part on the right side has a dimension of energy, but the left side N(z) is claimed to be count rate per time integration per altitude bin. This equation is not acceptable for publication. Furthermore, beta ($\beta$) is commonly used to represent volume backscatter coefficient, not backscattering cross-section, as the cross-section symbol is usually sigma ($\sigma$). Authors are suggested to consult with a commonly referenced class lecture at the following link, and use the more commonly accepted lidar equations and symbols.

http://superlidar.colorado.edu/Classes/Lidar2016/Lidar2016_Lecture04_LidarEquation.pdf

2) Page 9: Please provide a reference to Turkey Quartile test, as this isn't a common practice for most lidar people. BTW, it should be "when the signal to noise ratio approaches 1".

3) Page 12: Please provide a reference to the "one sided non-parametric Mann-Whitney-Wilcoxon rank-sum test" as it isn't common for the lidar field. BTW, what does "a scan" mean in Figure 6? Did you mean one profile?

4) Page 14, how are Si and Ni determined? Please provide a bit more details. Do you do this (equation (2)) for every altitude bin?

5) Page 16, notations are needed for equation (3).

6) Page 16, after the quadratic fit to the background, how do you handle such background and data? Did you mean to subtract the quadratic fitted background from the raw data? In this case, how do you handle the noise term in calculating SNR? Are photon counts still obey Poisson distribution? Please clarify in the manuscript.

7) Page 20, Figure 11, it's necessary to point out in the manuscript that satellite data aren't the real references as various satellites have their own calibration issues. Rayleigh temperatures around 90 km should be compared with ground-based resonance Doppler or Boltzmann lidar temperatures as these resonance lidars have much better signal to noise ratios at these altitudes.

8) Page 22-23, what do you mean by "misaligned"? A lidar beam was misaligned relative to its own receiver's field of view, or else? How were two lidars misaligned? Authors' writings here are confusing.

Minor comments on English writing:

As this is a very long paper, I strongly encourage authors go over the manuscript carefully to correct grammar and typo issues. For example, on page 24, near line 495, it should be "to initialize the inversion", not "initialized".

The paper title doesn't have good English grammar, for which I suggest to change "a 20 year cross-validation" to "a 20-year cross validation".

Please also note the supplement to this comment:
https://www.atmos-meas-tech-discuss.net/amt-2018-133/amt-2018-133-RC1-supplement.pdf
* * *

---

## Referee Comment (RC2) · Anonymous Referee #2 · 4 Jun 2018

The paper by Wing et al. is the first of a pair of papers describing the improvements made for Rayleigh lidar temperature retrieval and the utilization of this data set for comparison with satellite data. The paper is mostly well written and extensive. It covers general topics of Rayleigh temperature calculation and is therefore very important for the growing community of middle atmosphere lidars. After a repetition of the general design of the lidars used for this study, the authors describe potential issues for the data quality, like electronic signal contaminations, tilted background levels etc. A thorough examination of these issues is important and highly welcome. The cleaned photon

count profiles are used for the calculation of temperature profiles. Here, two co-located lidars allow for a comparison of the results and the removal of data with minor quality. The authors state that their procedures provide an improvement compared to the Rayleigh temperature calculations in the NDACC database. As a result, a data set for the calibration of satellite data is build.

As mentioned, the paper covers several important topics for Rayleigh lidar data. On the other hand, partly basic textbook knowledge is repeated. I recommend a more concise presentation of the study. Some topics are mentioned without physical reasoning, circumventing the transfer to other lidar systems. Several examples and additional aspects are given below. I recommend the revision of the manuscript, addressing these issues.

Specific comments:

- P6-7L140-146: In this section saturation is neglected, but in Section 3.5.1 the correction is described and in Figures 10, 13, 14, 15 stratospheric data is shown. I suggest not to neglect saturation throughout the manuscript.

- P7L147-150: I have not found any number on the integration time for the temperature profiles used here. I assume that it is long enough to at least partly overcome the issue of non-LTE. If not, the potential errors by assuming LTE need to be described. The statement "unable to relax" would not be sufficient, if differences between data sets are examined and "standards" are defined.

- P7L151: This assumption is problematic as there are different studies showing aerosols up to at least 35 km.

- P8Figure3: The count rates are comparatively high and saturation is likely to become a problem (see above). I assume a typo in the vertical resolution of 7.5 m.

- P9Sec3.3.1: Please explain the (potential) origin of these spikes. Fig. 4 shows that they easily reach 10-100 counts, i.e. they are quite substantial. I wonder whether

it would be useful to work on the origin of the spikes instead of only removing the resulting counts. Do you remove only the spiky bins or the whole profile? I guess, the first would result in too low counts rates in the altitude of the spikes after integration of several profiles. Please make clear.

- P10L207: Please explain "downstream counting rate".

- P10L207-219: Is the Kurtosis test always only done on the first 100 bins? If yes, how do you detect TES that may appear above that range? If no, how does the exponentially decreasing (i.e. non-Gaussian) signal influence the test

- P11L230-233: I do not see four groups of signals. Essentially it is either high background and low signal or low background and high signal. Please explain. Why does the number of groups depend on the statistics "the authors choose to use"? Which statistics? I am generally missing an explanation of the strategy or method. Why not simply defining a signal-noise-limit to separate good profiles and noisy profiles?

- P11L236 and Fig.6: The green line is not only a running average but contains some offset. Please explain.

- P11L238-243: It remains open how the blue line is derived. It is the result of a blackbox-software and the results are discarded by the authors. I suggest removing this section and the blue line in Fig. 6.

- P12L249-256 and Fig. 7: Please explain in more detail how this test works. Please use for Figure 7 the same data set as for Figure 6. Otherwise the reader can hardly comprehend the method. If I understand the test correctly, it only removes the worst profiles of a particular night. If the whole night has a bad signal, the data will not be removed. Correct? In line 253 you do "not exclude" the bad profiles, but in line 256 13 bad profiles are identified (how??) and "discarded". I do see a contradiction between the two sentences.

- P13Sec3.3.4: I am sorry, but I do not understand this section. Why not simply considering only data up to altitude z by defining a criterion like SNR(z) > Threshold ?

- P14L284-294: The noise reduction is interesting. To allow the reader evaluating the technical progress, it would be helpful to learn a) whether these are the most important changes in background count rate for the whole 20 y data set and b) what are the benefits for the temperature calculation if the background is reduced to 1/100 (e.g., range extended by .. km).

- P16Sec3.5.3: It would be helpful for the interpretation of the results (also of the companion paper) to have a quantitative description of the influence of a wrong background shape on the temperature calculation. Additionally, the SIN of the low channel in Fig. 9 is extremely high and the choice of the shape of the SIN profile is essential. Why quadratic?? I suggest validating the resulting temperature profile with independent information.

- P19Fig10: From my point of view the upper range of the temperature is somewhat optimistic. There seem to be superadiabatic gradients at 75 and 80 km. 30% relative error is ~70 K, i.e. the content of information is rather low. Which altitude is chosen for initialization? How is the signal smoothed for the choice of the initialization altitude (L395)? The melding of the signals should be visible in the uncertainties, but is not in Fig. 10. Please explain.

- P20Fig11: I suggest showing the error of the mean instead of the variance.

- P22L450: How many nights are excluded here?

- P22L452: Please mention the averaging window.

- P22L460: This conclusion cannot be proven without acknowledgement of the temperature uncertainty. The shaded area in Fig. 14 seems to show geophysical variability rather than measurement uncertainties. Fig. 13 shows persistent red or blue patches, indicating systematic differences between the lidars.

- P22Fig15: I am surprised about the small differences. Averaging the purple and blue

(40 and 50 km) line in Fig. 14, I would guess the difference is ∼1K. At 70 km the difference is close to 0 K, but ∼1 K in Fig. 11 (green and orange line). Is there any mis-interpretation from my side?

Minor comments:

P2L24-26: Please check this sentence (grammar).

P2L30-35: Please clean up the brackets, making this section easier to read.

P2L54: Remove "of two co-located lidar systems" and similar repetitions.

P3L56-61: I do not see this section relevant for the paper.

P5L99: I assume a dispersion of 0.3 mm/nm. Correct?

P6L136: "multiple scattered photons"

P6L137: "outside of the field of view"

P7L164-166: Example for textbook knowledge that can be removed.

P8L167-173: This section is partly redundant and should be shortened or removed.

P11L220: I suggest using "profile" instead of "scan".

P11L230: The intuition is always subjective. Please rephrase.

P11L235-236: I suggest deleting this sentence.

P13L260: Please explain "partial scan".

P18L367: "in an area of low signal"

P20L423: I suggest writing "The present study" instead of "This study".

P23L471: "colder" should read "lower"

P26L540-544: Sentences are mixed up. Please correct.

---

## Referee Comment (RC3) · Anonymous Referee #3 · 4 Jun 2018

The paper by Wing et al. is the first study to compare systematically long-term observations of two co-located Rayleigh lidar systems operating at 532 nm and 355 nm wavelengths. The author's motivation is the demonstration of the robustness and accuracy of their retrieved lidar temperatures and advertisement as a reference data set for validation of satellite measurements. The authors start with a description of both lidar systems followed by an extensive discussion of potential quality issues and data processing steps to resolve these issues, specifically electrical interference, signal induced noise, and saturation effects. Furthermore, the authors developed a robust scheme to

reject low quality lidar observations, resulting in higher signal-to-noise ratios and thus improved temperature data sets. In the last part the authors compare observations obtained by both instruments and, based on their finding of small systematic differences (<0.6 K), conclude that their lidar temperatures are accurate and thus suitable for satellite validation.

The paper is mostly well written and covers most aspects of the data processing which is typical for Rayleigh lidars. Some parts may be only relevant to older lidar systems, e.g. the detection of signal induced noise and electronic spikes, as newer systems generally do not suffer from these problems. While quality control is an important topic which is often neglected in publications, the authors also repeat basic knowledge commonly found in textbooks. I believe the manuscript can be made more concise by focusing on the important steps in data processing. Furthermore, in my opinion, differences in temperatures obtained by the two lidars should be analyzed in more detail. How do the differences (nightly menans) relate to uncertainties of the temperature profiles? E.g. are large differences visible in Figure 13 associated with large temperature uncertainties in one or both profiles? How are differences distributed?

I recommend the manuscript for revision.

Specific comments:

Please state the temporal resolution of your retrieved temperature profiles.

What does LiO3S stand for? In the paper you use different terms for the Ozone DIAL, e.g. line 86: OHP Differential Absorption Lidar, line 104: OHP DIAL, caption of Figure 2: LiO3S DIAL, line 162: LiO3S. Please use a single term to avoid confusion.

Figure 2: How are the 4 fibers combined before the chopper? Is the light coming off the four fibers coupled into a single fiber which relays the light to the receiver? What is the diameter of that fiber?

Line 153: You may add that this assumption excludes observations at mid to high

latitudes in summer where NLC can occur.

Section 2.3.2: Maybe more efforts should be spent on finding and eliminating the root cause of these transients. I believe there are lidar systems in operation which do not suffer from these problems.

Line 201: "We can see that the 22nd and 46th scans are contaminated by a TES with a duration of about 0.5 $\mu$s" I can't see that. The plot you are referring to is labeled with bins rather than time. What is a "scan"?

Line 215: "The kurtosis test is done in the time dimension as well as with altitude to exclude false positives in the photon count rate skew which may be due to clouds or aerosols." Well, a cirrus cloud drifting through the lidar beam might actually look like peaks in Figure 5.

Section 3.3.3: Since you do not precisely explain what the Matlab Neural Code does and how the blue trace is derived, I suggest you remove that part and shorten this section.

Line 234: "We have shown two approaches for attempting to address the issue..." Which approaches are you referring to? Do you mean the two approaches you explain below?

Lines 244-246: "The simple reality of ground based observation means that lidar signals clearly detect changes in the viewing conditions such as moonrise, thin cirrus clouds, optically thick clouds, changing light pollution, as well as changes in signal quality." What do you mean by "changes in signal quality"? I believe all aspects you listed, e.g. cirrus clouds, impact signal quality.

Figure 7: Why did you chose a different data set and not use the same data set shown in Figure 6? In order for the reader to evaluate the different algorithms, it would be beneficial to show results based on the same data set.

Linens 267-269: What is the reason for choosing "the point where the signal to noise

equals one in the density profile"? In equation (2), which profile do you use for determining the altitude of this point, the summed profile or the individual profile?

Figure 8: I am not sure about the unit on the y-axis. Shouldn't that be just Hz?

Section 3.5.1: What is the maximum count rate at which gating (or the chopper) cuts the profile off? Have you checked the validity of equation (3) within that range? One possibility would be plotting the correction as function of count rate using equation (3) and the actual measurements (ratio of high gain and low gain channels).

Lines 318-319: You can actually check the validity of this assumption using the 532 nm and 607 nm channels.

Section 3.5.3: You should not attempt to "correct" signal induced noise. It is fundamentally impossible to characterize properly signal induced noise in lidar signals because the noise is superposed on the atmospheric signal. Determining the signal induced noise from the background signal above the lidar signal is bound to fail because you are essentially observing the noise at different times outside the period where you actually are interested in. Signal induced noise is highly non-linear and therefore it is impossible to properly correct it. The data should be regarded as corrupt and not be used in lidar analysis. Besides, significant signal induced noise (e.g. blue trace in Figure 9) indicates that detectors are operated outside safe limits or there is a general technical problem with the lidar. If you insist on using the questionable data, you should assess how the retrieved temperature profile changes when you tweak your model representing the signal induced noise (e.g. cubic versus linear). How do your retrieved profiles compare to independent observations e.g. radiosondes at lower altitudes?

Figure 10: The superadiabatic gradient at approximately 75 km altitude looks suspicious to me. I assume the upper part of the profile is dominated by noise and initialization of the retrieval should happen at lower altitudes. Are 15 km removed from the top of the profile as indicated in line 92? Which altitude was chosen for initialization? Furthermore, I would expect the temperature uncertainties to increase where the transition from the upper to the lower channels happens. Please explain why this is not the case.

Figure 11: What is the shaded area?

Line 446: What do you mean by "mis-aligned"? Please explain.

Figure 13: It is hard to estimate absolute temperature differences. I suggest you use a segmented color bar with 6-10 different colors.

Can you provide a plot showing combined temperature error estimates of both lidar data sets? There is a period in mid 2001 with distinct blue color (negative temperature differences) between 30 and 55 km altitude. Could these observations also have been affected by misalignment? A similar area can be found in right after the last marked region in 2011.

Line 441: For clarification, the observation period of one lidar could be up to 20% longer compared to the other lidar? Why not just make both observation periods equal in length by cutting the longer observation?

Line 442: What is meant by "good internal alignment"?

Figure 14: Can you please mark periods of misalignment similar to Figure 13.

Lines 459-461: "Without excluding misaligned periods the lidar temperature differences are not significant as a function of altitude or year at the 2 sigma level". I am not sure if I understood that sentence correctly. Is the implication removal of misaligned periods causes the differences become significant intended?

Lines 462-463: "After removing comparisons between mis-aligned instruments we can calculate the ensemble median difference between the two systems." I do not understand that sentence. What was removed? The data affected by misalignment?

Line 486: "lidar measurements are accurate" I do not think you have sufficiently backed up this claim. Maybe it depends on what we understand by "accurate". I agree that the

long-term average (20 years) appears to be accurate, however according to Figure 13 nightly means obtained by the two co-located lidars can differ by more than 10 K. What is the reason for these large differences? Are these large differences expected from an SNR point of view, or are there other maybe unknown error sources which average out on long time scales?

—————————————————————

---

## Author Comment (AC1) · 16 Jul 2018

Dear Referee, Thank you very much for your helpful comments and suggestions. I have attempted to address each of your concerns to the best of my ability. If you would like me to implement further changes or iterations on a point please let me know. I appreciate your efforts to help me improve this paper.

The questions you raise about error estimation after removal of signal induced noise contributions are in my opinion very critical. I think that as a lidar community we really

need to push the envelope on our data retrieval techniques and error estimates - particularly if we want to do new work in the Mesosphere/Thermosphere. The work in this paper is not perfect but it is an improvement to the commonly used (Hauchecorne/Chanin 1980) lidar temperature inversion. It's my belief that as a community we should continue investigating improvements to our techniques. A fully Bayesian approach such as the Optimal Estimation Technique presented by (Sica/Haefele 2015) might be a profitable endeavor. As an added benefit a Bayesian Technique produces full averaging kernels which would make lidar data much more attractive for assimilation to people in the satellite and reanalysis communities .
* * *
Response Lidar temperature series in the middle atmosphere as a reference data set. Part A: Improved retrievals and a 20 year cross-validation of two co-located French lidars: Referee #1
* * *
1) Page 6: Lidar equation (1) has dimension mismatch on the left and right hand sides. The first part on the right side has a dimension of energy, but the left side N(z) is claimed to be count rate per time integration per altitude bin. This equation is not acceptable for publication. Furthermore, beta ($\beta$) is commonly used to represent volume backscatter coefficient, not backscattering cross-section, as the cross-section symbol is usually sigma ($\sigma$). Authors are suggested to consult with a commonly referenced class lecture at the following link, and use the more commonly accepted lidar equations and symbols.

Good catch thank you. I've changed divided the right hand side by the photon energy hc/lambda and changed $\beta$\_cross to $\sigma$\_cross
* * *
2) Page 9: Please provide a reference to Turkey Quartile test, as this isn't a common practice for most lidar people. BTW, it should be "when the signal to noise ratio approaches 1".

Cited Tukey(1949) "signal to noise" changed to "signal to noise ratio"

\*\*\*\*\*\*\*\*\*\*\*\*\*\*\*\*\*\*\*\*

3) Page 12: Please provide a reference to the "one sided non-parametric MannWhitney-Wilcoxon rank-sum test" as it isn't common for the lidar field. BTW, what does "a scan" mean in Figure 6? Did you mean one profile?

Cited Mann and Whitney (1947)

I've always called a single level_0 or level_1 photon count time series a 'scan' and used the word 'profile' for level_2 things like density, pressure, temperature. Reviewer #2 and Reviewer #3 made the same point so perhaps it's a personal idiosyncrasy. In any case, I've changed all occurrences of 'scan' to 'profile'

\*\*\*\*\*\*\*\*\*\*\*\*\*\*\*\*\*\*\*\*

4) Page 14, how are Si and Ni determined? Please provide a bit more details. Do you do this (equation (2)) for every altitude bin?

Line 268 - 272 The noise is always evaluated between 120 km and 155 km and the altitude range for the evaluating the signal is defined as the scale height below the point where the signal to noise equals one in the density profile. Each individual profile has a value representing the signal, $S_{i}$, and a noise, $N_{i}$. The profile values are compared to the nightly sum of the signal, $S_{sum}$ and the nightly sum of the noise, $N_{sum}$. Changed to: The noise of an individual profile, $N_{i}$, is expressed as the summation of photon counts in bins which fall between 120 km and 155 km and the nightly noise, $N_{sum}$ is the summation of all $N_{i}$ for the night. To determine a metric for the nightly average lidar signal, $S_{sum}$, we first calculate a quick density profile and determine the lowest altitude where the signal to noise ratio equals 1. Then we calculate the altitude that is one density scale height ($\sim$8 km) below this point.

The lidar range bins which correspond to this altitude range are then summed to yield $S_{sum}$. A similar calculation, using the same range bins as in the nightly average calculation, is done to determine the signal of single profile, $S_{i}$.
* * *
5) Page 16, notations are needed for equation (3).

$N$, $\tau$, and $\Delta t$ are described in lines 309-310. We have now replaced the definition of $N$ with separate definitions for $N_{counted}$ and $N_{received}$, as they appear in Equation (3): Replacement text: The background theory and derivation of Eq. (3) is well described by (Donovan et al., 1993), where $N_{received}$ is the number of photons incident on the PMT per measurement time interval and $N_{counted}$ is the number of photons per measurement time interval which are actually counted by the system. In general, $N_{counted}$ < $N_{received}$ due to effects of the system deadtime.
* * *
6) Page 16, after the quadratic fit to the background, how do you handle such background and data? Did you mean to subtract the quadratic fitted background from the raw data? In this case, how do you handle the noise term in calculating SNR? Are photon counts still obey Poisson distribution? Please clarify in the manuscript.

Yes, in the case of a quadratic background I subtract the quadratic function from the entire photon counts profile in exactly the same way I would treat a constant or linear background.

As you correctly point out, as soon as there is signal induced noise the profile is no longer Poisson as the count rate in each lidar bin is no longer fully independent of the count rates in the bins on either side of it. The Total counts are some combination of 'Real counts' and 'Contamination counts ' (T = R + C) with a common shot noise dT = 1/sqrt(T) with some contribution dC = 1/sqrt(C) coming from the Signal Induced

Noise portion and dR = 1/sqrt(R) representing the noise from all other sources. When I'm using the linear or quadratic backgrounds I am making an assumption that I'm completely removing the signal induced noise, C and I no longer have to add dR and dC in quadrature. I'm approximating dN ∼= dR and that the photon count profiles are now approximately Poisson.

On page 16 line 334, we have added a new sentence to the manuscript: "...as our estimate of signal induced noise. The best background function is subtracted from the raw photon counts profile."

On page 16 line 341 we have added a new sentence to clarify about SNR: "...than the simple quadratic approximation. For the quadratic case, as soon as there is signal induced noise the profiles no longer represent Poisson distributions as the count rate in each lidar bin is no longer fully independent of the count rates in the bins on either side of it. Therefore, precise calculations of the SNR would require the addition in quadrature of real noise (from sky background and signal photon counts) and contamination noise (from signal induced noise). Here, however, we make the assumption that the signal induced noise is able to be completely removed from the raw profiles with the subtraction of the quadratic function. We therefore interpret the background subtracted profiles to obey approximately Poisson distributions, thereby approximating the total noise in the profile to the noise of only the real photons, which can be treated as uncorrelated."
* * *
7) Page 20, Figure 11, it's necessary to point out in the manuscript that satellite data aren't the real references as various satellites have their own calibration issues. Rayleigh temperatures around 90 km should be compared with ground-based resonance Doppler or Boltzmann lidar temperatures as these resonance lidars have much better signal to noise ratios at these altitudes.

Line 414 inserted text: It is important to note that additional complications exist when

comparing temperatures derived from ground based lidars to temperatures derived from satellite data which have their own calibration concerns. We explore the issues of lidar-satellite comparison in Part B of this paper. A co-located ground-based resonance Doppler or Boltzmann lidar would provide a better comparison data set as resonance lidars have high signal to noise ratios above 85 km (Alpers, 2004).
* * *
8) Page 22-23, what do you mean by "misaligned"? A lidar beam was misaligned relative to its own receiver's field of view, or else? How were two lidars misaligned? Authors' writings here are confusing.

In both lidar systems the high gain Rayleigh channel has 4 mirrors, each of which needs to be aligned independently with respect to the laser in the sky and also the fibre optic with respect to the primary focus of the mirror. In LTA the low gain channel is a single independent mirror. So a total of 9 mirrors need to be aligned every night to make Rayleigh measurements.

Line 456 inserted text: Internal misalignments happen when one or more of the five mirrors in LTA or four mirrors in LiO3S is not properly aligned with the laser or the fibre optic is not centered on the focal point of the mirror.
* * *
Minor comments on English writing: As this is a very long paper, I strongly encourage authors go over the manuscript carefully to correct grammar and typo issues. For example, on page 24, near line 495, it should be "to initialize the inversion", not "initialized". The paper title doesn't have good English grammar, for which I suggest to change "a 20 year cross-validation" to "a 20-year cross validation

I will have an anglophone colleague read over the paper for grammatical errors.

Please also note the supplement to this comment:

https://www.atmos-meas-tech-discuss.net/amt-2018-133/amt-2018-133-AC1-supplement.pdf

---

## Author Comment (AC2) · 16 Jul 2018

Dear Referee, Thank you very much for your helpful comments and suggestions. I have attempted to address each of your concerns to the best of my ability. If you would like me to implement further changes or iterations on a point please let me know. I appreciate your efforts to help me improve this paper.

Thank you for raising the point about aerosols. It forced me to do quite a bit of reading on the subject. I think this is a very important question for lidars that measure

both Rayleigh temperatures and Mie scatter from aerosols. (Hauchecorne and Chanin 1980) is given very often in lidar papers to justify the 30 km assumption of a clean atmosphere. I think this is valid most of the time but I can see problems in some of the old temperature profiles after the 1982 El Chichon eruption. It really underscores the need for a Raman channel.

Thanks also for helping me to clarify my language when describing the statistics and my discussion of LTE. I think this work is more clear after removing some of my idiosyncratic language.
* * *
Response Lidar temperature series in the middle atmosphere as a reference data set. Part A: Improved retrievals and a 20 year cross-validation of two co-located French lidars: Referee #2 Specific comments: P6-7L140-146: In this section saturation is neglected, but in Section 3.5.1 the correction is described and in Figures 10, 13, 14, 15 stratospheric data is shown. I suggest not to neglect saturation throughout the manuscript.

Text added line 146: A correction for saturation in the lower stratosphere is described in Sect. \ref{Deadtime Correction}
* * *
P7L147-150: I have not found any number on the integration time for the temperature profiles used here. I assume that it is long enough to at least partly overcome the issue of non-LTE. If not, the potential errors by assuming LTE need to be described. The statement "unable to relax" would not be sufficient, if differences between data sets are examined and "standards" are defined.

Changed In this work we are unable to relax this assumption. To However, given that a single lidar profile is acquired every 2.8 minutes and a nightly average temperature is generated every 4 hours, we can have some confidence in this assumption.
* * *
P7L151: This assumption is problematic as there are different studies showing aerosols up to at least 35 km.

You are correct. In the presence of significant aerosol loading (volcanos and fires) we can see a cold bias in our temperatures above 30 km. However, in times when the aerosol loading is less pronounced the Rayleigh lidar temperature cold bias is relatively small and can generally be corrected by using the Raman lidar channel. I've weakened my assumption and provided two justifications for my assumptions. Changed assumption 4 and citation to read:

Fourth, we assume that the atmosphere at mid-latitudes is generally free of aerosols above 30 km when there are no active volcanic or fire events (Hauchecorne and Chanin, 1980). During less severe background aerosol conditions (aerosol scattering ratio < 1.02), (Gross et al. 1997) suggests lidar temperature cold biases due to Mie scattering are less than 0.5 K at 20 km.

(Khaykin et al. 2017) published a 22 year stratospheric aerosol climatology for OHP https://www.atmos-chem-phys.net/17/1829/2017/acp-17-1829-2017.html See reponse figure 1

They found that during quiescent times (no major eruptions or fires) that the scattering ratio was near one at 30 km

Here is a plot of temperature from our recent ozone blind NDACC validation campaign at OHP. The two OHP temperatures from LTA (532 nm) and LiO3S (355 nm) are in good agreement with the NASA mobile validation lidar (355 nm) above 30 km. N.B. that this particular LTA profile doesn't have a 607 nm Raman correction as the Raman channel experienced some difficulties during this period. But even without a Raman correction the temperatures converge by 30 km. See reponse figure 2
* * *
P8Figure3: The count rates are comparatively high and saturation is likely to become a problem (see above). I assume a typo in the vertical resolution of 7.5 m.

Typo corrected to 75 m
* * *
P9Sec3.3.1: Please explain the (potential) origin of these spikes. Fig. 4 shows that they easily reach 10-100 counts, i.e. they are quite substantial. I wonder whether it would be useful to work on the origin of the spikes instead of only removing the resulting counts. Do you remove only the spiky bins or the whole profile? I guess, the first would result in too low counts rates in the altitude of the spikes after integration of several profiles. Please make clear. The spikes can be expected to occur in any Poisson counting process, could be induced by some thermal or electronic imperfection in the photomultiplier, small charges in the Licel digital recorder, interaction of the photocathode substrate with a cosmic ray, or dozens of different kinds of electronic 'cross-talk' between all the instruments at the observatory station.

On any given night there are three lidars, several ozone instruments, OH spectrometers, and probably several other instruments at various times in simultaneous operation at the Geophysical Observatory building at OHP. I would genuinely love to track down every little bug and glitch in the lidar system but unfortunately, I'm in my 3rd year of a 3 year PhD in Paris. The observatory is in Haute Provence in the south of France and only CNRS technicians are permitted to make changes to the experiment. I don't have the time or access required to address this problem at the experimental level. Additionally, this project takes advantage of measurements taken as many as 20 years ago. While increasing the quality of all future lidar data is indeed a positive goal for the lidar group, all existing data requires some software treatment in any case. Therefore, I've done my best to address the spikes with software - and this approach appears to be both adequate and successful. Individual spiky bins are removed from the profiles. When averaging multiple profiles together, it is possible to do so in a manner which

accounts for bins with "not a number" (i.e. spiky bins whose data is totally removed) separately from bins which have "zero counts" (bins which have zero photons, but which are still valid data). For example, the nanmean function in Matlab. The overall SNR may decrease slightly at the altitude of the spiky bin (since we're adding bins from fewer profiles into the average, which is equivalent to taking fewer measurements at that altitude), but the averaged count rate will not be skewed too low.

Caption Fig04 has been clarified to say "Tukey Quartile spike identification based on the signal difference between consecutive lidar time bins for short integration lidar returns. An entire night of lidar profiles is over-plotted in the stack plot. The black line is the 2 sigma limit and points above this line are removed.".

We have added a few sentences to page 9 line 189: "...and inaccurate background estimations. The spikes can have many potential origins (thermal or electronic imperfection in the photomultiplier, small charges in the Licel digital recorder, interaction of the photocathode substrate with a cosmic ray, or dozens of different kinds of electronic 'cross-talk' between all the instruments at the observatory station) and are therefore impossible, in practical terms, to completely prevent in the lidar data set, and completely impossible to prevent in measurements which have already been made. Therefore, it is necessary to address this problem using software during the analysis."
* * *
P10L207: Please explain "downstream counting rate".

"downstream counting rate" changed to "counting rate in bins subsequent to the TES burst"
* * *
P10L207-219: Is the Kurtosis test always only done on the first 100 bins? If yes, how do you detect TES that may appear above that range? If no, how does the exponentially decreasing (i.e. non-Gaussian) signal influence the test

Changed the Fig5 caption to read, with 2 extra sentences for clarity: Figure 5: Upper panel is a surface plot of lidar returns as a function of time bin and scan number. For clarity, only the first 100 bins are shown in this plot. The test is carried out using all bins of each profile. Two instances of TES can be seen as anomalous peaks in the photon count rate. Lower panel is a summation of the fourth statistical moment (kurtosis/skew) for each scan. The red line indicates a 2\sigma limit on the skew of the population. Points above the limit are excluded.
* * *
P11L230-233: I do not see four groups of signals. Essentially it is either high background and low signal or low background and high signal. Please explain. Why does the number of groups depend on the statistics "the authors choose to use"? Which statistics? I am generally missing an explanation of the strategy or method. Why not simply defining a signal-noise-limit to separate good profiles and noisy profiles?

This is what I'm trying to show. You say 2 groups of signals and that is very reasonable. I say that the median of the first period of low signal and high noise is significantly different from the second and they are in fact two different populations. Whether that depends on laser power, sky transmission, background or something else - I don't know. That's why I'm trying to develop an automated tool for data quality assessment. We have changed the sentences on line 229-233 to read:

However, when we look at the panel representing the signal, it is equally reasonable to, instead, interpret the plot as containing four groups. Each of these groups has similar signals which match fairly well with the changes in the backgrounds shown in the panels above (profiles 1-23, profiles 24-35, profiles 36 - 48 and profiles 49 - 92) . However, whether these four groups of signals should be treated in analysis as two, three, or four distinct populations is open to interpretation. Therefore, we seek an objective programmatic solution for identifying bad scans.

In essence I have defined a signal to noise cut off, as you suggest, with equation (2),

in section 3.3.4 about "Good Scans". This is indeed useful for identifying and rejecting scans which contribute more noise than information to the nightly average at a given altitude. Therefore, it is the final quality control step used.

However, before we get to that point, we address in section 3.3.3 about "Bad Scans" the separate question of rejecting scans which do not conform to the general population, and are therefore outliers. We point out on line 246 that there can be multiple signal to noise population medians during the course of the night, which makes setting a constant minimum SNR criterion for the whole night inappropriate as a sole means of judging good vs. bad scans. The one sided non-parametric Mann-Whitney-Wilcoxon rank-sum test, is the solution to the subjective interpretation problems presented by Figure 6. It is not the final step in quality control - but it is a useful intermediate step.
* * *
P11L236 and Fig.6: The green line is not only a running average but contains some offset. Please explain.

Fig06 Caption has been clarified: Example of lidar signal and noise during a night of measurements. Top panel shows the total background counts summed from 120 km to 153 km and the bottom panel shows the total signal summed between 35 km and 40 km. Green bounds are calculated based on a smoothed 2$\sigma$ error estimation of the summed photon counts (red) and the blue line is an attempt to estimate local population medians using the Matlab Neural Network tool.
* * *
P11L238-243: It remains open how the blue line is derived. It is the result of a blackbox-software and the results are discarded by the authors. I suggest removing this section and the blue line in Fig. 6.

I've had several discussions at NDACC lidar meetings and at the last IRLC about using machine learning and using MatLab's neural network toolbox to estimate lidar profile

backgrounds. I think that is plot will be interesting to several people. Using machine learning to process lidar data is to my knowledge a wide open field to be explored. Your point is well taken about blackbox-software.
* * *
P12L249-256 and Fig. 7: Please explain in more detail how this test works. Please use for Figure 7 the same data set as for Figure 6. Otherwise the reader can hardly comprehend the method. If I understand the test correctly, it only removes the worst profiles of a particular night. If the whole night has a bad signal, the data will not be removed. Correct? In line 253 you do "not exclude" the bad profiles, but in line 256 13 bad profiles are identified (how??) and "discarded". I do see a contradiction between the two sentences.

Thanks for catching the mistake between Fig 6 and Fig 7. Figure 6 has been remade over the same range as Figure 7.

Cited Mann and Whitney (1947) for details of the statistic

This Mann-Whitney-Wilcoxon test only removes profiles which are "very different" compared to others nearby on the same night, and you are correct that it will not remove all profiles if the whole night has bad signal. The latter is not its purpose. The Mann-Whitney-Wilcoxon test has two ways in which scans are determined to be outliers: (a) SNR is too low compared to that of nearby scans and (b) SNR is too high compared to that of nearby scans. To apply this test to lidar, we want to reject from our analysis scans which fail for reason (a; scan is low quality), but not those which fail for reason (b; scan is high quality). Therefore the last sentences are not contradictory: We in fact have not rejected any scans on the basis of high quality ("failure reason (b)"), but have rejected 13 scans on the basis of low quality ("failure reason (a)"). To address data for which the whole night has very bad signal: First, OHP operators monitor the lidar measurements as they are made throughout the night, and attempt to either correct the issue or stop the measurement if the sky clouds in. The first 'quality filter' is the

judgement of the OHP technicians. Second, I have an arbitrary condition that an LTA temperature profile must reach 80 km in 4 hours integration at 2 km effective vertical resolution. This catches the few remaining 'bad nights' where the lidar acquisition was too short or the observing conditions were too cloudy. The OHP operators are very good at maintaining data integrity.

\*\*\*\*\*\*\*\*\*\*\*\*\*\*\*\*\*\*\*\*\*\*\*\*\*\*\*\*\*\*

P13Sec3.3.4: I am sorry, but I do not understand this section. Why not simply considering only data up to altitude z by defining a criterion like SNR(z) > Threshold ?

Because there is no flexibility in that kind of SNR definition. I used 5,676 nights of lidar data from two instruments over 20 years. I needed something that could be adaptable to changing signal levels as transmitter power changes over decades. As well I wanted to use the data as efficiently as possible. On a clear night I can get temperatures up to 90 km but there's no point in wasting a night of data with light cirrus where I only get temperatures up to 80 km.

\*\*\*\*\*\*\*\*\*\*\*\*\*\*\*\*\*\*\*\*\*\*\*\*\*\*\*\*\*\*

P14L284-294: The noise reduction is interesting. To allow the reader evaluating the technical progress, it would be helpful to learn a) whether these are the most important changes in background count rate for the whole 20 y data set and b) what are the benefits for the temperature calculation if the background is reduced to 1/100 (e.g., range extended by .. km).

I agree. This is an area of lidar science that is waiting to be developed with the aid of modern computers and new models. As mentioned previously I have been in communication with colleagues looking to use Bayesian statistics and machine learning to look at lidar backgrounds and noise. This paper is already very long and I'm in the 3rd year of a 3 year PhD. Perhaps this work could be done in a different article?

\*\*\*\*\*\*\*\*\*\*\*\*\*\*\*\*\*\*\*\*\*\*\*\*\*\*\*\*\*\*

P16Sec3.5.3: It would be helpful for the interpretation of the results (also of the companion paper) to have a quantitative description of the influence of a wrong background shape on the temperature calculation. Additionally, the SIN of the low channel in Fig. 9 is extremely high and the choice of the shape of the SIN profile is essential. Why quadratic?? I suggest validating the resulting temperature profile with independent information.

I think a full quantitative description of background is going to require another article. Combined with your previous point I see the outline of a very interesting project. Thanks for the great questions more work is definitely required in this area.

I tried both exponential fits and splines to model the SIN but neither were very stable solutions. Given small changes in my background selection or fitting parameters the exponential changed too drastically. I used a quadratic because it is better than a linear fit, gives me stable and reproducible results (which were important for processing so many nights of data), and removes most of the SIN in the region where the signal to noise ratio is close to one. This is not a perfect solution but, I think it is an incremental improvement.

To my knowledge there are no independent validation sources that are appropriate. I think that two co-located lidars are the best we can hope for. Satellites have their own calibration issues as I point out in part B.
* * *
P19Fig10: From my point of view the upper range of the temperature is somewhat optimistic. There seem to be superadiabatic gradients at 75 and 80 km. 30% relative error is âĹij70 K, i.e. the content of information is rather low. Which altitude is chosen for initialization? How is the signal smoothed for the choice of the initialization altitude (L395)? The melding of the signals should be visible in the uncertainties, but is not in Fig. 10. Please explain.

Please note that this example temperature profile was calculated at 300 m vertical resolution. I was simply demonstrating a troposphere to mesosphere temperature profile at high vertical resolution. The relative error drops significantly at 1 km vertical resolution and we generally get 30% error above 90 km.

I use a 3rd order Savitzky-Golay filter with a small 11 point window. This filtering is not passed though into the data product it is only applied to the photon counts profile for the purpose of determining where the lowest altitude where the signal to noise ratio is equal to one. This altitude is different every night and depends on the transmitter power, nightly integration time, and sky conditions.

I use a relative error weighting function to minimize the total uncertainty. This ensures that I'm not adding extra noise to my photon counts and makes the transition in the temperature profile as smooth as possible.
* * *
P20Fig11: I suggest showing the error of the mean instead of the variance.

Error on the median added. Caption and text updated.
* * *
P22L450: How many nights are excluded here?

I set a 2 sided p-value of 0.05. I didn't think to record the number of nights excluded just my confidence interval.
* * *
P22L452: Please mention the averaging window.

Added to text: 'A 30 day averaging window is applied to each of the four curves.'
* * *
P22L460: This conclusion cannot be proven without acknowledgement of the temperature uncertainty. The shaded area in Fig. 14 seems to show geophysical variability rather than measurement uncertainties. Fig. 13 shows persistent red or blue patches, indicating systematic differences between the lidars.

Good point! Thanks. I've added the following text: 'For reference, a typical LTA temperature profile with an effective vertical resolution of 2 km has an uncertainty due to statistical error of 0.3 K at 40 km; 0.7 K at 50 km; 1.5 K at 60 km; and 4.6 K at 70 km.' The two lidars are measuring the same air at the same time, so it can't be geophysical variability.

Yes. There are time periods where there appears to be internal misalignment in one or the other lidar. This is most definitely a systematic error. That's why I tried to identify and remove these time periods using the Chi squared method before plotting Fig15. I think this is a very good technical argument to be made for investing in automated alignment systems in lidars. To my knowledge this is the first time that anyone has looked at comparing co-located lidar signals over such a long period. I think it shows that perhaps the lidar community needs to do further work on testing signal linearity and overlap corrections.

\*\*\*\*\*\*\*\*\*\*\*\*\*\*\*\*\*\*\*\*\*\*\*\*\*\*\*\*\*\*\*

P22Fig15: I am surprised about the small differences. Averaging the purple and blue (40 and 50 km) line in Fig. 14, I would guess the difference is âĹij1K. At 70 km the difference is close to 0 K, but âĹij1 K in Fig. 11 (green and orange line). Is there any mis-interpretation from my side?

Going back to my response to your comment on the variance in Fig11 this is an example of the error on the median. It looks really small given the large amount of data being considered. So no misunderstanding on your part - given all 20 years of data there is a (un)remarkable degree of consistency between two co-located lidars.

\*\*\*\*\*\*\*\*\*\*\*\*\*\*\*\*\*\*\*\*\*\*\*\*\*\*\*\*\*\*\*

Minor comments: P2L24-26: Please check this sentence (grammar).

Inserted "and"

\*\*\*\*\*\*\*\*\*\*\*\*\*\*\*\*\*\*\*\*\*\*\*\*\*\*\*\*\*\*

P2L30-35: Please clean up the brackets, making this section easier to read.

Changed to square braces

\*\*\*\*\*\*\*\*\*\*\*\*\*\*\*\*\*\*\*\*\*\*\*\*\*\*\*\*\*\*

P2L54: Remove "of two co-located lidar systems" and similar repetitions.

I have removed some of the repeated redundant wording. However, there are 3 to 5 independant co-located lidars at OHP (depending on how you count them). This work only uses two of the OHP lidars.

\*\*\*\*\*\*\*\*\*\*\*\*\*\*\*\*\*\*\*\*\*\*\*\*\*\*\*\*\*\*

P3L56-61: I do not see this section relevant for the paper.

Motivation for other DIAL systems to submit validated temperature profiles to NDACC. I think that the people are hesitant to put forward 355 temperatures for validation as NDACC data products when the main focus of their system is ozone.

\*\*\*\*\*\*\*\*\*\*\*\*\*\*\*\*\*\*\*\*\*\*\*\*\*\*\*\*\*\*

P5L99: I assume a dispersion of 0.3 mm/nm. Correct?

Typo corrected

\*\*\*\*\*\*\*\*\*\*\*\*\*\*\*\*\*\*\*\*\*\*\*\*\*\*\*\*\*\*

P6L136: "multiple scattered photons"

We mean photons which have each scattered multiple times. We do not mean multiple photons which have each been scattered. We could change to "multiply-scattered

photons". Please advise.

I will check with an anglophone. But I think this is correct.
* * *
P6L137: "outside of the field of view"

Done
* * *
P7L164-166: Example for textbook knowledge that can be removed.

This is intuitive to lidar scientists familiar with remote sensing but the prompt can be useful for modelers, satellite scientists, and pure geophysicists as an orientation point.
* * *
P8L167-173: This section is partly redundant and should be shortened or removed.

The authors felt it was important to show a co-added lidar signal as this may not be obvious outside our community.
* * *
P11L220: I suggest using "profile" instead of "scan".

Done
* * *
P11L230: The intuition is always subjective. Please rephrase.

We have addressed this comment in our changes to the previous comment for P11L230-233, in which this and adjacent sentences have been reworked.
* * *
P11L235-236: I suggest deleting this sentence.

Done

\*\*\*\*\*\*\*\*\*\*\*\*\*\*\*\*\*\*\*\*\*\*\*\*\*\*\*\*\*\*

P13L260: Please explain "partial scan".

Changed to "partial profile". Using a partial profile entails only using the linear portions of the photon count time series and cutting out instead of correcting saturation, spikes, and other data problems.

\*\*\*\*\*\*\*\*\*\*\*\*\*\*\*\*\*\*\*\*\*\*\*\*\*\*\*\*\*\*

P18L367: "in an area of low signal"

Changed "area" to "region"

\*\*\*\*\*\*\*\*\*\*\*\*\*\*\*\*\*\*\*\*\*\*\*\*\*\*\*\*\*\*

P20L423: I suggest writing "The present study" instead of "This study".

Done

\*\*\*\*\*\*\*\*\*\*\*\*\*\*\*\*\*\*\*\*\*\*\*\*\*\*\*\*\*\*

P23L471: "colder" should read "lower"

Done

\*\*\*\*\*\*\*\*\*\*\*\*\*\*\*\*\*\*\*\*\*\*\*\*\*\*\*\*\*\*

P26L540-544: Sentences are mixed up. Please correct.

We have edited this section so far as possible, given the limitations of proper names, in two languages, of the various funding agencies.

This section has been corrected to read:

"Acknowledgements. The data used in this paper were obtained as part of the Network for the Detection of Atmospheric Composition Change (NDACC) and are publicly available (see http://www.ndacc.org, http://cdsespri. ipsl.fr/NDACC) as well as from the SABER (see ftp://saber.gats-inc.com) and MLS (see https://mls.jpl.nasa.gov) data centres for public access. This work is supported by the Atmospheric dynamics Research InfraStructure Project (ARISE 2) which is funded by the European Union's Horizon 2020 research and innovation programme under grant agreement No. 653980. French NDACC activities are supported by Institut National des Sciences de l'Univers/Centre National de la Recherche Scientifique (INSU/CNRS), Université de Versailles Saint-Quentin-en-Yvelines (UVSQ), and Centre National d'Études Spatiales (CNES). The authors would also like to thank the technicians at La Station Géophysique Gérard Mégie at OHP.

Please also note the supplement to this comment:
https://www.atmos-meas-tech-discuss.net/amt-2018-133/amt-2018-133-AC2-supplement.pdf

[Figure]

(a) OHP LiO3S, SR at 532 nm, 1994–2015 quiescent

**Fig. 1.**

**Fig. 2.**

---

## Author Comment (AC3) · 16 Jul 2018

Response Lidar temperature series in the middle atmosphere as a reference data set. Part A: Improved retrievals and a 20 year cross-validation of two co-located French lidars: Referee #3

Dear Referee, Thank you very much for your helpful comments and suggestions. I have attempted to address each of your concerns to the best of my ability. If you would like me to implement further changes or iterations on a point please let me know. I

appreciate your efforts to help me improve this paper.

I appreciate the push back you have given on Signal Induced Noise corrections. These are exactly the kind of discussions we need to have when designing new lidar experiments. As well we need to be aware of these kinds of problems in older lidar systems as we can't change the past. The scientific value of the OHP lidar database is often forgotten. This year marks the 40th year of continuous lidar measurements of temperature between 30 km and 85 km. This data record is longer and more stable than any satellite or rocketsonde record. Creating good software tools to deal with noise is essential for getting the most out of this exceptional resource.

You're point about lidar temperature 'accuracy' was also particularly thought provoking. I think that temperatures in the middle atmosphere are often dismissed as a 'solved problem' however, Part B of this paper shows fundamental disagreements between the lidar and satellites on the 'simple' question of stratopause heights. There may still be some unresolved questions surrounding the 'true state' of the atmosphere and how well we can know it though different measurement techniques.
* * *
Specific comments:
* * *
Please state the temporal resolution of your retrieved temperature profiles.

Inserted P6_L170: ' However, given that a single lidar profile is acquired every 2.8 minutes and a nightly average temperature is generated every 4 hours'
* * *
What does LiO3S stand for? In the paper you use different terms for the Ozone DIAL, e.g. line 86: OHP Differential Absorption Lidar, line 104: OHP DIAL, caption of Figure 2: LiO3S DIAL, line 162: LiO3S. Please use a single term to avoid confusion.

Added P4_L86 "also referred to as Lidar Ozone Stratosphère (LiO3S)"
* * *
Figure 2: How are the 4 fibers combined before the chopper? Is the light coming off the four fibers coupled into a single fiber which relays the light to the receiver? What is the diameter of that fiber?

They're arranged in a linear fashion using a commercial fibre optic bundler. Diameter of the bundle is 1600 um.
* * *
Line 153: You may add that this assumption excludes observations at mid to high latitudes in summer where NLC can occur.

Sightings of NLCs over OHP seem to be extremely rare. (Pérot et al. 2010) First climatology of polar mesospheric clouds from GOMOS/ENVISAT stellar occultation instrument https://www.atmos-chem-phys.net/10/2723/2010/ has seen only a few NLCs over decades of measurements.
* * *
Section 2.3.2: Maybe more efforts should be spent on finding and eliminating the root cause of these transients. I believe there are lidar systems in operation which do not suffer from these problems.

I didn't know these phenomena existed before looking at a full 20 years of lidar profiles. It is equally plausible that other lidar systems also have these problems but are unaware of them. I'm a PhD student in my third and final year and I can't justify a weeks long trip to the south of France to investigate these issues. I have tried to correct the issue to the best of my ability in the software. I spoke at some length with the representatives from Licel at the IRLC2018 conference and they showed some level of concern over these TES. However, given that they do not occur frequently or with regularity it is

difficult to track down.

\*\*\*\*\*\*\*\*\*\*\*\*\*\*\*\*\*\*\*\*\*\*\*\*\*\*\*\*\*\*

Line 201: "We can see that the 22nd and 46th scans are contaminated by a TES with a duration of about 0.5 $\mu$s" I can't see that. The plot you are referring to is labeled with bins rather than time. What is a "scan"?

Each bin is 0.1 $\mu$s and the FWHM is about 5 bins. The exact width is less important than the fact that there is a temporal duration to this signal. This is what differentiates a TES from a spike which occurs in a single isolated bin without affecting data in adjacent bins.

Changed 'scan' to 'profile' everywhere in the text. These profiles are 1.6 minutes long, each as indicated on line 210.

Changed the sentence on page 10 line 208 in the text to read: "In the example shown in Fig. 5 (top) is a surface plot of counts differences between consecutive altitude bins for the first 100 altitude bins of lidar data. Each bin is 0.1 $\mu$s wide . "

Changed first sentence of Fig 5 caption to read: "Figure 5: Upper panel is a surface plot of lidar returns as a function of altitude bin and profile number...."

\*\*\*\*\*\*\*\*\*\*\*\*\*\*\*\*\*\*\*\*\*\*\*\*\*\*\*\*\*\*

Line 215: "The kurtosis test is done in the time dimension as well as with altitude to exclude false positives in the photon count rate skew which may be due to clouds or aerosols." Well, a cirrus cloud drifting through the lidar beam might actually look like peaks in Figure 5.

Yes. That's why I do it in the profiles as well. I use a simple test to see if a potential TES shows up at the same altitude for multiple profiles. If it does it's a cloud not a TES.

\*\*\*\*\*\*\*\*\*\*\*\*\*\*\*\*\*\*\*\*\*\*\*\*\*\*\*\*\*\*
Section 3.3.3: Since you do not precisely explain what the Matlab Neural Code does and how the blue trace is derived, I suggest you remove that part and shorten this section.

I have had several discussions at NDACC lidar meetings and at the last IRLC about using machine learning and MatLab's neural network toolbox to estimate lidar profile backgrounds. I think that is plot will be interesting to several people.
* * *
Line 234: "We have shown two approaches for attempting to address the issue. . ." Which approaches are you referring to? Do you mean the two approaches you explain below?

Tense is changed and sentence clarified
* * *
Lines 244-246: "The simple reality of ground based observation means that lidar signals clearly detect changes in the viewing conditions such as moonrise, thin cirrus clouds, optically thick clouds, changing light pollution, as well as changes in signal quality." What do you mean by "changes in signal quality"? I believe all aspects you listed, e.g. cirrus clouds, impact signal quality.

The simple reality of ground based observation means that lidar signals clearly detect changes in the viewing conditions such as moonrise, thin cirrus clouds, optically thick clouds, changing light pollution, as well as changes in signal quality due to operational instrumental factors including laser power, data acquisition noise, cleanliness of optics, and issues with signal saturation.
* * *
Figure 7: Why did you chose a different data set and not use the same data set shown in Figure 6? In order for the reader to evaluate the different algorithms, it would be

[Figure]

beneficial to show results based on the same data set.

Thanks for catching the mistake. I've remade Fig06 over the same range as Fig07.
* * *
Linens 267-269: What is the reason for choosing "the point where the signal to noise equals one in the density profile"?

It seems like a reasonable choice for deciding on an arbitrary starting point. We were motivated by getting temperatures in the UMLT. However, the point is well taken. I'm aware that other groups use other definitions. It would be good to see a study devoted specifically on this topic.
* * *
In equation (2), which profile do you use for determining the altitude of this point, the summed profile or the individual profile?

Nightly sum.
* * *
Figure 8: I am not sure about the unit on the y-axis. Shouldn't that be just Hz?

Changed
* * *
Section 3.5.1: What is the maximum count rate at which gating (or the chopper) cuts the profile off? Have you checked the validity of equation (3) within that range? One possibility would be plotting the correction as function of count rate using equation (3) and the actual measurements (ratio of high gain and low gain channels).

The gating is based on an adjustable time delay not maximum count rate. The High gain Raleigh channel is currently blanked at 22 km and the low gain Rayleigh channel is blanked at 12 km.
We have added the following text on page 15 after line 312 to clarify. In order to measure the deadtime experimentally, we assume that the low gain channel, because it has low photon count rates, will always operate in the linear response regime and will never suffer from deadtime effects. Thus, it represents a value proportional to the "true" rate for returned photons for each altitude. Once scaled by a constant (e.g. using MSIS or another model), we can use this count rate as N_received. The high gain channel, conversely, measures higher photon count rates at every altitude than the low gain channel does. Similarly to the low gain channel, at the low end of its dynamic range, the high gain channel operates linearly, and therefore represents a value proportional to the "true" rate for returned photons for each altitude. The constant of proportionality is different for low and high gain channels. At low count rates, the scaled counts measured by the high gain and low gain channels are equal. As photon count rates move into the higher end of the high gain channel's dynamic range, deadtime begins to have an effect: The high gain channel will measure too few photons compared to the "true" rate; the number of photons which are returned to the lidar. Therefore, we call the scaled high gain count rate N_uncorrected in equation 3; it has not yet been dead time corrected. We will refer to the deadtime corrected scaled high gain count rate as N_dtc.

Equation 3 is used several times. First, we use data only from altitudes for which the low gain and high gain channels both have measurements (nominally X to X km). We iterate through various values of \tau, increasing by an increment each time, calculating a N_dtc for each N_uncorrected value. This is carried out until the difference between N_corrected (from the high gain channel) and N_received (from the low gain channel) is minimized. This determines the dead time of the system, \tau.

Next, equation 3 is used again, using the measured nightly value for \tau, to calculate N_dtc for all N_uncorrected high gain channel measurements. This allows us to correct the high gain measurements for the entire profile.

Variable names in Eq(3) have also been changed for clarity

No I haven't investigated the validity of Eq(3). I'm relying on the work of (Donovan 1993). However, if I iterate through N_recieved and N_counted I get a reasonable convergence

No. We use the high and low gain Rayleigh channels as well as the nitrogen Raman channel to directly measure the correction. The set values are only used if the data is unavailable. If the data is unavailable we have no reference to compare to.
* * *
Lines 318-319: You can actually check the validity of this assumption using the 532 nm and 607 nm channels.

Yes more recent data uses 607 nm channel to make the correction. Unfortunately, the N2 Raman channel was not present for the entire period from 1993.
* * *
Section 3.5.3: You should not attempt to "correct" signal induced noise. It is fundamentally impossible to characterize properly signal induced noise in lidar signals because the noise is superposed on the atmospheric signal. Determining the signal induced noise from the background signal above the lidar signal is bound to fail because you are essentially observing the noise at different times outside the period where you actually are interested in. Signal induced noise is highly non-linear and therefore it is impossible to properly correct it. The data should be regarded as corrupt and not be used in lidar analysis. Besides, significant signal induced noise (e.g. blue trace in Figure 9) indicates that detectors are operated outside safe limits or there is a general technical problem with the lidar. If you insist on using the questionable data, you should assess how the retrieved temperature profile changes when you tweak your model representing the signal induced noise (e.g. cubic versus linear). How do your retrieved profiles compare to independent observations e.g. radiosondes at lower altitudes?

I disagree with the conclusion that we should not make the attempt at a SIN correction. You are quite right that a perfect correction might be impossible. However, we have found that a correction of the sort described in the paper, for the types of signal induced noise that we see at OHP, can be adequately applied for the purposes of our temperature retrievals. The effects of this signal induced noise in our profiles, when uncorrected, is to warm the upper altitude regions of the temperature profiles. Conveniently, we have two measurement channels (the high and low gain channels) which make coincident measurements in this region. Typical count rates within this region are are well within the linear response regime of the high gain channel; therefore dead time correction is not required at these altitudes, and we can believe the high gain channel temperature profile in this region. The quadratic correction for signal induced noise in the low gain channel brings the resulting low gain temperatures into agreement with those from the high gain channel at these high altitudes.

While it would be wonderful to eliminate every stray source of noise in the lidar, we cannot do this for the measurements going back 40 years and more - which form a valuable data set. We also point out that the effect of this quadratically-characterized signal induced noise is negligible at low altitudes: For example, in Fig. 9, the SIN contribution at 30 km is less than 100 counts, compared to a bg + signal value in the tens of MHz (see fig03). In terms of contribution to temperature, this is so small as to not be observable.

I did some initial quality testing between my 3 channel lidar temperature retrieval and the radiosondes launched from the station at Nimes (∼150 km west) and the results are reasonable. There's some expected differences but the results can be very good when the sonde travels directly east. That said the focus of this paper is above 30 km and a full radiosonde comparison study with calculated air mass trajectories would be a good project for the next student.
* * *
Figure 10: The superadiabatic gradient at approximately 75 km altitude looks suspi-

cious to me. I assume the upper part of the profile is dominated by noise a sition from the upper to the lower channels happens. Please explain why this is not the case.

Please note that this example temperature profile was calculated at 300 m vertical resolution. I was simply demonstrating a troposphere to mesosphere temperature profile at high vertical resolution. The relative error drops significantly at 1 km vertical resolution and we generally get 30% error above 90 km.

No. The low gain Rayleigh channel contributes nothing that high up. When I meld the two channels I weight the addition by the relative error.
* * *
Figure 11: What is the shaded area?

Now noted in caption it's the ensemble variance.
* * *
Line 446: What do you mean by "mis-aligned"? Please explain.

In both lidar systems the high gain Rayleigh channel has 4 mirrors each of which needs to be aligned independently. In LTA the low gain channel is a single independent mirror. So a total of 9 mirrors need to be aligned every night to make a Rayleigh measurement.

Line 456 inserted text: Internal misalignments happen when one or more of the five mirrors in LTA or four mirrors in LiO3S is not properly aligned with the laser or the fibre optic is not centered on the focal point of the mirror.
* * *
Figure 13: It is hard to estimate absolute temperature differences. I suggest you use a segmented color bar with 6-10 different colors. Can you provide a plot showing combined temperature error estimates of both lidar data sets? There is a period in mid 2001 with distinct blue color (negative temperature differences) between 30 and 55 km

altitude. Could these observations also have been affected by misalignment? A similar area can be found in right after the last marked region in 2011.

The same information is already presented in a more compact way in Fig14

I've added the following text: 'For reference, a typical LTA temperature profile with an effective vertical resolution of 2 km has an uncertainty due to statistical error of 0.2 K at 40 km; 0.4 K at 50 km; 0.6 K at 60 km; 0.7 K at 70 km; 1.8 K at 80 km; and 602 K at 90 km. For reference, a typical LiO3S temperature profile with an effective vertical resolution of 2 km has an uncertainty due to statistical error of 0.3 K at 40 km; 0.5 K at 50 km; 1.0 K at 60 km; 2.7 K at 70 km; and 10 K at 80 km.'

I cannot account for the blue regions in Fig13 based on either lidar uncertainty budget or through geophysical explanations. Yes you're correct the blue bias between 30-50 km is likely due to misalignment. Given 5 mirrors in LTA and 4 mirrors in LiO3S there are many possible ways to be misaligned. As well the severity of the misalignment

\*\*\*\*\*\*\*\*\*\*\*\*\*\*\*\*\*\*\*\*\*\*\*\*\*\*\*\*\*\*\*\*

Line 441: For clarification, the observation period of one lidar could be up to 20% longer compared to the other lidar? Why not just make both observation periods equal in length by cutting the longer observation?

4 hours is the standard OHP temperature measurement. This criterion excludes the few cases when there is a significant temporal offset between the two lidars. Maybe one lidar was being temperamental and took an extra hour to start up. I didn't look into why some measurements were not coincident I just excluded them if they were too different.

\*\*\*\*\*\*\*\*\*\*\*\*\*\*\*\*\*\*\*\*\*\*\*\*\*\*\*\*\*\*\*\*

Line 442: What is meant by "good internal alignment"? Figure 14: Can you please mark periods of misalignment similar to Figure 13.

In both lidar systems the high gain Rayleigh channel has 4 mirrors each of which needs to be aligned independently. In LTA the low gain channel is a single independent mirror. So a total of 9 mirrors need to be aligned every night to make a Rayleigh measurement.

Line 456 inserted text: Internal misalignments happen when one or more of the five mirrors in LTA or four mirrors in LiO3S is not properly aligned with the laser or the fibre optic is not centered on the focal point of the mirror.

Fig14 is marked
* * *
Lines 459-461: "Without excluding misaligned periods the lidar temperature differences are not significant as a function of altitude or year at the 2 sigma level". I am not sure if I understood that sentence correctly. Is the implication removal of misaligned periods causes the differences become significant intended?

Thanks for picking up on that implication - indeed it is not what we mean. We have rephrased the sentence.

Replace lines 459 through 461 with this text: Using all data, including misaligned periods (example: winter 2006-2007 in Fig. 13 and Fig. 14) none of the lidar temperature differences are significant at the 2-sigma level, although certain periods do have temperature differences which are detectable at the 1-sigma level. This can be seen where the blue shaded region (2005 - 2008) and the magenta shaded region (in 2007) are entirely above the zero line. If the misaligned periods are disregarded, no temperature differences are significant, even at the 1-sigma level. Therefore, we conclude that the results from the lidars, when well-aligned, are stable in time, over the 20-year period studied.
* * *
Lines 462-463: "After removing comparisons between mis-aligned instruments we can calculate the ensemble median difference between the two systems." I do not under-

stand that sentence. What was removed? The data affected by misalignment?

Yes. A chi-squared test was used to detect these nights and exclude them from the rest of the analysis.
* * *
Line 486: "lidar measurements are accurate" I do not think you have sufficiently backed up this claim. Maybe it depends on what we understand by "accurate". I agree that the long-term average (20 years) appears to be accurate, however according to Figure 13 nightly means obtained by the two co-located lidars can differ by more than 10 K. What is the reason for these large differences? Are these large differences expected from an SNR point of view, or are there other maybe unknown error sources which average out on long time scales?

I completely agree the differences cannot be explained from a SNR point of view. I think that alignment is a major unaddressed problem in lidar science. With a single lidar we often assume that our transmitter and receiver are well aligned if we can maximise the count rates at some reference altitude. Before conducting the comparison between the two lidar systems I did not fully realize how great of an effect slight misalignment can have on the resulting temperature profile.

When we make temperature comparisons with other techniques like radiosondes, satellites, or other sensors we can easily dismiss small deviations based on claims geophysical variability, sampling slightly different air masses, averaging effects, or sampling error. However, in this study we have two active remote sensors, making measurements in the same building, at the same resolution, operated by the same technicians, designed by the same optical and electrical engineers, and compared over 2 decades. They should be the same. But since they are not I think it is legitimate to entertain the possibility that manual nightly alignment of lidars is not as robust or repeatable as we like to assume.

The Pandora's box that is lidar alignment is really horrifying when you sit and try to imagine all the possible sources of alignment drift: operator change blindness (not noticing small changes over a long period of time), thermal changes shifting optics, optical degradation, hysteresis in optical mounts, angular sensitivity of optics which exceeds manufacturer specifications. Etc.

Your point about 'accurate' is well taken. Changed to 'reasonable'

Please also note the supplement to this comment:
https://www.atmos-meas-tech-discuss.net/amt-2018-133/amt-2018-133-AC3-supplement.pdf

[Figure]

**Fig. 1.**

---

## Author Response (AR2)

**This is from Referee #3**

Thank you for providing additional information and clarifications. Though most of my concerns were addressed adequately in your response, some issues still remain. I wrote additional comments concerning those issues below. Where relevant, I reproduced my original remarks/questions and your responses.
* * *
Original comment: Section 3.3.3: Since you do not precisely explain what the Matlab Neural Code does and how the blue trace is derived, I suggest you remove that part and shorten this section.

Author: I have had several discussions at NDACC lidar meetings and at the last IRLC about using machine learning and MatLab's neural network toolbox to estimate lidar profile backgrounds. I think that is plot will be interesting to several people.

Reviewer: I do not question potential interest within the community in machine learning and using Matlab's toolbox. Machine learning is a complex and much hyped topic, but many issues regarding e.g. reproducibility of the results are not yet well understood, as results often depend on how the particular network was trained and what training data were used.

If you do not provide any information about how you set up the neural network and how it was trained, nobody will ever be able to reproduce your findings or compare with other results. Thus, publishing those results will be worthless for the science community.

You may show results obtained with the Matlab Neural Network tool in your paper. However, you should briefly explain what you did. Just saying "I used this toolbox and that is what I got" is no scientific work. On the other hand, you could say "I took the Fourier transform of time series x and here is the spectrum", as the Fourier transform is a well-defined algorithm which will always produce the same results when applied to the same data sets. With neural networks many things can go wrong because their behavior is not so well-defined. In your case that is especially important given your statement "the software requires an exhaustive set of example bad profiles which we cannot supply". So how was the network trained without a representative training data set? Did you use any trick nobody else knows?

Coming up with a precise description of what the neural network did will likely be difficult. For that reason, and because the neural network part is not essential for your paper's main results, I suggested to remove this part. However, you may insist on showing these results. In that case I will insist on a description of the neural network. That description may be brief, but all essential information which allows someone to repeat your steps must be given.

**Discussion of Matlab's neural networking tool are removed.  Figure06 is edited.**
* * *
Line 245: "bad scans" -> "bad profiles"?

**Changed**
* * *
Original comment: Lines 267-269: What is the reason for choosing "the point where the signal to noise equals one in the density profile"?

Author: It seems like a reasonable choice for deciding on an arbitrary starting point. We were motivated by getting temperatures in the UMLT. However, the point is well taken. I'm aware that other groups use other definitions. It would be good to see a study devoted specifically on this topic.

Reviewer: I may sound harsh, but your answer "it seems like a reasonable choice" does not convey any scientific value. To be more precise, why is that a reasonable choice? If you did not investigate this question, it is ok to say so (given the limited time, no study can be perfect). I am aware of different groups using different definitions for the starting point, and I was just curious whether you have any convincing arguments for a particular definition.

**We will be more precise in describing the quantitative motivations for our choice of SNR=1.  I did some basic testing of different SNR values 3 years ago, and found that the results were a compromise between starting the temperature inversion at an altitude which is too high, where the profile contained more noise than signal (SNR<1; resulted in variability at the top of the temperature profiles which is not geophysical) and starting the inversion at an altitude which is too low (SNR=5 for example; resulted in having to start the temperature inversion a few km lower, and with the Hauchecorne-Chanin retrieval requiring yet further altitude to be cut off of the retrieved temperature profiles in the region before the uncertainty converges to acceptable values, we lost significant information in the altitude range we wish to study). Therefore, SNR=1 was chosen, as it is the least restrictive criterion for batch processing our temperatures which still ensured that the temperature inversions were initialized more by information than by noise.**

**We have added the following text to line 269:**
**"We chose a cutoff value of SNR=1 because it is the least strict value we could use which ensures that we have more information than noise (or, specifically, more information than**

**noise plus background counts), at the altitude within the density profile where we begin the downward temperature integration. Had we chosen a criterion which was less strict (SNR<<1), we would expect to see more statistical variability in the top altitudes of the temperature retrieval as a result of starting the temperature integration in a region which contains more noise than signal. Conversely, choosing a criterion which is too strict (SNR>>1) limits the maximum altitude of the temperature retrieval as discussed in section 3.6.1."**
* * *
Original comment: Section 3.5.3: You should not attempt to "correct" signal induced noise. It is fundamentally impossible to characterize properly signal induced noise in lidar signals because the noise is superposed on the atmospheric signal. Determining the signal induced noise from the background signal above the lidar signal is bound to fail because you are essentially observing the noise at different times outside the period where you actually are interested in. Signal induced noise is highly non-linear and therefore it is impossible to properly correct it. The data should be regarded as corrupt and not be used in lidar analysis. Besides, significant signal induced noise (e.g. blue trace in Figure 9) indicates that detectors are operated outside safe limits or there is a general technical problem with the lidar. If you insist on using the questionable data, you should assess how the retrieved temperature profile changes when you tweak your model representing the signal induced noise (e.g. cubic versus linear). How do your retrieved profiles compare to independent observations e.g. radiosondes at lower altitudes?

Author: I disagree with the conclusion that we should not make the attempt at a SIN correction. You are quite right that a perfect correction might be impossible. However, we have found that a correction of the sort described in the paper, for the types of signal induced noise that we see at OHP, can be adequately applied for the purposes of our temperature retrievals. The effects of this signal induced noise in our profiles, when uncorrected, is to warm the upper altitude regions of the temperature profiles. Conveniently, we have two measurement channels (the high and low gain channels) which make coincident measurements in this region. Typical count rates within this region are are well within the linear response regime of the high gain channel; therefore dead time correction is not required at these altitudes, and we can believe the high gain channel temperature profile in this region. The quadratic correction for signal induced noise in the low gain channel brings the resulting low gain temperatures into agreement with those from the high gain channel at these high altitudes.

While it would be wonderful to eliminate every stray source of noise in the lidar, we cannot do this for the measurements going back 40 years and more - which form a valuable data set. We also point out that the effect of this quadratically-characterized signal induced noise is negligible at low altitudes: For example, in Fig. 9, the SIN contribution at 30 km is less than 100 counts,

compared to a bg + signal value in the tens of MHz (see fig03). In terms of contribution to temperature, this is so small as to not be observable.

I did some initial quality testing between my 3 channel lidar temperature retrieval and the radiosondes launched from the station at Nimes (~150 km west) and the results are reasonable. There's some expected differences but the results can be very good when the sonde travels directly east. That said the focus of this paper is above 30 km and a full radiosonde comparison study with calculated air mass trajectories would be a good project for the next student.

Reviewer: I agree, we can't change the past and need to work with the data at hand. Above you mention the agreement between high and low channel temperatures, which improves when the quadratic correction is applied. That is very valuable information, because it gives credibility to your approach, and should be mentioned in your manuscript as well. On the other hand, your validation works only for the lower channel. In the absence of any validation for the upper channel which shows a completely different behavior (linear versus cubic), you could at least provide an estimate of the magnitude of the correction, e.g. x K at 75 km, where x is the difference between temperature profiles retrieved with and without correction. If x is sufficiently large (e.g. >1 K), that should be acknowledged as potential source of error, as signal induced noise is a dynamic phenomenon which commonly depends on several factors (e.g. peak intensity, average intensity, particular type of detector) and thus likely varies over a broad range of time scales (from pulse-to-pulse to months). The problem is that signal induced noise causes a non-Gaussian error, so integrating longer does not help you. Your correction most likely helps alleviating the problem, but it won't be perfect. How well it really works – we don't know. E.g. you may unknowingly overcorrect the noise resulting in a cold bias, or undercorrect and still retain a warm bias. Only the comparison with an independent data set can tell whether your correction is working as it is supposed to. However, if your x is small (I think it is. Unless both lidar systems show exactly the same behavior, I would expect larger differences in Fig. 15 for a large x.), you may argue that the effect of signal induced noise on temperature is small as well. Because of the problems it causes, most groups try to avoid signal induced noise by limiting the peak count rate to safe levels.

**Added near pg17 line 379 (updated version of article):**
**"We have some confidence that the quadratic background correction to the low gain channel correctly approximates the moderate non-linear signal induced error because we can compare the corrected low gain channels to the high gain channel. In the overlap region we have two channels making coincident measurements and we can safely assume that the response rate for the high gain channel is linear. Therefore, a correction for signal induced noise in the low gain channel which brings the resulting low gain count rates into the closest agreement with the high gain channel count rates at the same altitudes will be**

the optimal choice for the correction. In some cases, the quadratic correction for signal induced noise in the low gain channel yields better agreement than the constant or linear corrections, in which case it is employed. The best individual choice (constant, linear, quadratic) is used for each profile. We believe these empirical corrections to be sufficient, because (a) the resulting agreement with the high gain channel improves as compared to the uncorrected profile, and (b) the resulting corrected low gain count profiles are generally equal to the high gain count profiles to within statistical uncertainty, and (c) for the few cases in which the empirical correction ultimately fails, this will be apparent by the corrected signal retaining poor SNR values. The melding procedures of Section 3.6 weight the combined high and low gain Rayleigh channels according to SNR, and so in these cases, the poorly-corrected low gain contributions to the final melded counts profile will be negligible, and all information will be obtained from the high gain channel."

For the high gain channel, there is no extra channel to compare to. Instead, we consider the shape of a standard idealized lidar profile which contains no signal induced noise: That which has only a constant background with no slope nor any curvature. We are confident that the majority of the high gain channel profiles exhibit this behaviour - the constant background is in fact the most commonly fit background for this channel. For the linear and quadratic backgrounds which are sometimes exhibited by this channel, we reason that any correction which can bring the corrected background into closer agreement with the shape of an idealized background (i.e. a constant about zero) will be optimal. The profiles for which we have applied a linear or quadratic background removal do indeed result in such corrected profiles and we deem this to be sufficient for the purposes of the current study.

The ultimate effects of a quadratic background correction on the retrieved temperature profile, as compared to a constant background correction for the same profile, are possible to calculate. However, it is not immediately apparent how to properly characterize the effect of the quadratic non-linear SIN correction as a function of altitude for profiles in general. The magnitude of the change in temperature depends both on the quality of the nightly integrated photon counts profile (a clear night with high signal will receive a correction at a different altitude than cloudy night with lower laser power), slope of the linear correction or the curvature of the quadratic correction, and the underlying 'true' photon counts gradient. Plus there are confounding variables, correcting the background changes the a priori initialisation altitude, the value of the a priori, the convergence of the temperature algorithm, and the choice of vertical integration scale. The effect of a quadratic background correction on the temperature.

**You're absolutely correct my correction is far from perfect. I've done my best to correctly identify and subtract the noise and background in the lidar profiles. In Figure 11 where I compare the results of the changes made in the algorithm to the satellite temperatures we see that the corrections work together to minimize the temperature differences. It may be interesting in future to delve into the relative temperature contributions of each of the corrections identified in this paper. For the current project, we ensured that each correction was an improvement within the arena in which we were testing it (i.e. removal of TES, we checked that TES were indeed removed from profiles. For removing background, we check that the resulting counts profile did not have any remaining structure at the background altitudes). We did not run the full temperature analysis with every combination of corrections turned on and off - although this is a parameter space we could certainly explore at some later date.**

**The peak count rate is well within safe limits. We cover the full dynamic range of photon returns by sharing the altitude range between two Rayleigh channels, each designed specifically to each cover only a part of the dynamic range. This keeps count rates on each PMT within the linear regime. As shown in Figure 3, there is no evidence of large nonlinear response in the raw counts profiles. If there was, we would expect to see a flattening of the profiles at the lower end of each altitude range, in addition to the flattening induced by the intentional electronic blanking of the PMTs. We do not see such an effect. Further, nonlinearity tests were carried out during the critical review of French lidars (REF: Kekhut 1993; section 5b), which determined the linear regime for each PMT, and the regime which is correctible via dead time correction (to within a 5% uncertainty). The count rates shown in Figure 3 of our paper are well within the linear regime of the counting hardware.**
* * *
Original comment: Figure 13: It is hard to estimate absolute temperature differences. I suggest you use a segmented color bar with 6-10 different colors. Can you provide a plot showing combined temperature error estimates of both lidar data sets? There is a period in mid 2001 with distinct blue color (negative temperature differences) between 30 and 55 km altitude. Could these observations also have been affected by misalignment? A similar area can be found in right after the last marked region in 2011.

Author: The same information is already presented in a more compact way in Fig14 I've added the following text: 'For reference, a typical LTA temperature profile with an effective vertical resolution of 2 km has an uncertainty due to statistical error of 0.2 K at 40 km; 0.4 K at 50 km; 0.6 K at 60 km; 0.7 K at 70 km; 1.8 K at 80 km; and 602 K at 90 km. For reference, a typical LiO3S temperature profile with an effective vertical resolution of 2 km has an uncertainty due to

statistical error of 0.3 K at 40 km; 0.5 K at 50 km; 1.0 K at 60 km; 2.7 K at 70 km; and 10 K at 80 km.' I cannot account for the blue regions in Fig13 based on either lidar uncertainty budget or through geophysical explanations. Yes you're correct the blue bias between 30-50 km is likely due to misalignment. Given 5 mirrors in LTA and 4 mirrors in LiO3S there are many possible ways to be misaligned. As well the severity of the misalignment.

Reviewer: If blue (or red) biases outside the boxes may be caused by misalignment, then misalignment is obviously a major source of error which ultimately limits the accuracy (and, depending on time scale, also precision) of your measurements.

**We agree. This is a limitation of our lidar system. Prior to this work, the effects of misalignment had not been identified in the long-term OHP data set. Therefore, identifying that this is a contributing factor to the OHP temperature analyses (as shown here), and making a first attempt to identify these regions programmatically in the data, is a first step to mitigating this factor. In future we will follow up with more sophisticated tests to address this important issue.**

**We have added the following text to acknowledge this in the manuscript, at line 450:**
**"It is possible that the criteria described above for identifying periods of misalignment is not yet stringent enough. Therefore, one limitation of the OHP measurements in terms of accuracy, and depending on time scale, also precision, is the influence of periods of misalignment that have not been programmatically identified. An ideal solution would be to have an independent method of monitoring mirror alignment during atmospheric measurements (e.g. installation of a small sighting telescope to measure the alignment coupled with an automatic fiber optic alignment system). With the existing data set from OHP extending back two decades, we unfortunately cannot retrospectively address such a hardware goal, but there may be opportunities in future to look into the effects of choosing different criteria to identify periods of misalignment."**

**This is from Referee #2**

This review is on the revised manuscript of the Wing et al. paper. The authors addressed the comments of the previous review in the extensive (appreciated!) reply and in the new manuscript. The paper is improved and much better comprehensible, now. Nevertheless, there are still some remarks, either to be repeated or new topics coming up with the new version.
* * *
The general challenge of the paper is the combination of different aspects (or goals of the paper) and their proper description in the text, that I see better now after reading the review reply. I find three goals: i) the introduction of the long-term data set and the instrumental changes, iii) treatment of this heterogeneous data set – or quality assurance - for the use in the accompanying paper, iii) improvement of the temperature algorithm and reduction of the bias compared to satellite soundings. Of course, these goals cannot be completely separated from each other, but they often affect different altitude sections. I recommend to make these goals clearer, try to subdivide the text appropriately or refer each (sub)section to its particular goals. As an example, the instrumental achievements for background reduction (Sec. 3.4) are dedicated to the first goal, but the deadtime correction (3.5.1) to the second.

The following text has been added to

**This work follows three main goals: i) the introduction of the long-term data set and the instrumental changes, ii) treatment of this heterogeneous data set for the use in the accompanying paper, and iii) improvement of the temperature algorithm and reduction of the bias compared to satellite soundings. These goals cannot be completely separated from each other, but goal i) is broadly addressed in sections 2.1-3.2 and 3.4, goal ii) is addressed in sections 3.3.2-3.4 and again in sections 3.5-4, goal iii) is addressed in section 5.**
* * *
Line 45 – end: Please refer "first part" (second, third) to the Section numbers used in the manuscript.

**Changed.**
* * *
l. 72: Please explain shortly how your Mie channel is working if it uses a similar filter as the Rayleigh channels.

**We have specified on line 72 that the Mie channel is 532 nm.**

**The Mie aerosol channel and Raman water vapour channel are not central to this work. A full discussion of OHP aerosol lidar can be seen in the following reference, which we have included at the end of section 2.1:**

Khaykin et al., *Variability and evolution of the midlatitude stratospheric aerosol budget from 22 years of ground-based lidar and satellite observations*, Atmospheric Chemistry and Physics, vol. 17, no. 3, pg. 1829--1845, 2017. 10.5194/acp-17-1829-2017
* * *
Table 1: Please check the dimension of the mirror diameters.

**'mm' changed to 'cm'**
* * *
Section 3.2: I got confused (and the general reader may also) about the altitude resolutions, integration times, and potential smoothing. I find raw data with 75 m resolution, temperate data with 300 m and 2 km, nightly means and individual profiles. I recommend showing raw data only at the resolution used for the quality control, and temperatures as nightly means with the altitude resolution used for the accompanying paper.

**We have presented plots at the relevant resolution for each step of the data processing procedures. Section 4 (re-written) clearly summarizes the resolutions used for each stage. The quality control steps are not all done at the same resolution.**

**Altitude resolutions used:**
- **Raw measurements are made 75 m. Figure 3 is plotted at 75 m.**
- **Spike and LTS corrections must be done at the native resolution, of 75 m. Figures 4 and 5 are plotted at 75 m.**
- **Further quality control can be done at the processing resolution of 300 m. Therefore, figures 6, 7, and 9 are plotted at 300 m resolution.**
- **The temperature profile is calculated at 300 m resolution. Figure 10 is plotted at 300 m.**
- **Next, for comparison with LiO3S and satellite measurements, the LTA NDACC temperature (black) results are integrated to a effective resolution of 2 km, while the LTA new algorithm (green) results remain at 300 m. Figure 11 is plotted at these resolutions.**
- **LTA new algorithm temperatures are then integrated to 2 km effective resolution for quantitative comparison with LiO3S values, which are at 2 km resolution. Figure 13 is plotted at 2 km resolution.**

**We have made a more detailed caption for Fig 10, as it is a nightly mean profile. We have now specified that clearly.**

**The integration times at OHP are set by hand each night. The times indicated in the plots are correct, although not necessarily constant from night to night.**
* * *
l. 184: I suggest writing "overestimation of the background due to localized signal contaminations" (noise should not be confused with background)

**Changed.**
* * *
l. 185: "warmer temperatures" should read "higher temperatures"

**Changed.**
* * *
l. 185-187: I suggest removing these sentences, because signal contaminations will normally not result in underestimation of the true background (detector noise, moonlight etc.). The sentences may be moved to Section 3.5.3.

**Removed:**
**The opposite holds true for an underestimation of the background (produces a colder profile).**
* * *
l. 240 to …: Please accept that the reader may not be able to identify three or more groups, but only "high background at low signal" and vice versa. Please explain in more detail or (preferred!) just state after introducing the red lines in Fig. 6 that this case may be simple but you are seeking for a flexible solution for 20 years of data (instrumental changes) and various conditions. Furthermore, express that you are searching for the outliers within a single night, not bad-signal profiles in general.

**We included the text in lines 243-244 to address exactly the first issue that you point out: It's quite likely that a given reader may not identify three or more groups, although other readers may do so. We have pointed out the particular profile number ranges for the groups we ourselves identify so that the reader can follow along with one example of a reasonable grouping, but there's no reason that someone else would choose the same groups. It is therefore preferable to have an objective, rather than a subjective, test to**

**determine outliers within a single night. The point you make is exactly one motivation for this test.**

**Our use of the term "programmatic solution" on line 245, and all text on lines 249-253 was intended to address the second comment: That we are seeking a solution which can be applied to all sorts of measurements we may acquire in various conditions.**

**To address the third comment, the first paragraph of section 3.3.3 has been rewritten to read:**
**"After the removal of lidar profiles which suffer from clear signal contamination, there may still be profiles which ought not be included in a lidar temperature analysis because they are outliers of poor quality compared to other profiles within the same night. Conceptually, 'bad profiles' are lidar profiles with a high background and/or a low signal strength as compared to profiles measured shortly before or after the profile in question. These profiles need to be positively identified as not belonging to the general population of nightly lidar profiles".**
* * *
l. 249-254: I still recommend to delete this section and the blue line. This is not a project report. I doubt that you want to state that the Matlab software cannot be used in general. Therefore this section may at most be interesting for the NDACC people you talked to, but the general reader will be confused.

**Done.**
* * *
l. 260-267: I am sorry but I still do not understand how the MWW rank-sum test is applied here. As far as I understand, the test checks which of two distributions is larger. Do you simply want to check whether the background is larger than the signal at 35-45 km? Then you may not need the cumulative sum. How the rank sum is calculated and how does this compare to the cumulative sum of either background or signal? Why the first 13 profiles are discarded? From my point of view this test is a central method of the quality control procedures and therefore should be clearly described.

**The first null hypothesis here is that all the lidar profiles belong to a single continuous distribution with a single nightly median. On a clear night with constant laser power and a relatively constant background we can easily fail to reject the null and we have no real need for a non-parametric statistic. However, in our example shown in Figure 6 and 7 we have too much signal and background variation in our example for this to be the case.**

Failing this criterion we employ the MWW rank sum test which allows the identification of outliers based on N sub-population medians with non-gaussian distribution. A regular T-test will not work here as the assumptions we need to make about the populations are not true. For example, anomalies in noise are most likely to be high while anomalies in signal are likely to be low but anomalies in SNR can go either way. As another example, consider the SNR responses for transient thin cirrus clouds vs. moonlight filtering through transient optically thick clouds.

By using the cumulative sum we can more easily identity significant changes in a signal. Look and figure 6 and figure 7 and mentally try to look for subpopulations. It's much easier to conceptualize the ordinally ranked and cumulatively summed data in figure 7 than to guess at divisions in figure 6. Cumulative sums are the natural choice of variable to use in a MWW test.

Here's a back of the envelope calculation just looking at Fig 7. Unfortunately, I don't save the intermediate statistical scores - I'm only interested in the end determination and whether I should include a profile or not.

Step 1: Do a cumulative sum for the signal and the noise as seen in Figure 7. Take SNR and arrange largest to smallest   This allows us to establish an ordinal rank for the data. Given that all lidar data is positive (no negative photon counts) this rank usually but not always corresponds to profile number. Exceptional cases of poor SNR will be assigned a low ordinal rank and the best SNR will have high ordinal rank.

Step 2: In each of the 4 sections of Figure 6 there are some higher and some lower SNR values. I want to see whether there are significant differences in the sub-population medians. The expectation value for the rank each sub-median is U_all = (number of observations in sub-median)*(total number of profiles +1)/2
Example  Profiles 1-13: U_1-13 = 13*(92+1)2 = 604.5

Step 3: Do a Z distribution score   Z =((sum of ranks) - (mean of population))/(E_w)
Where E_w is the standard error of the Wilcox sum. E_w = sqrt((number of observations in sub-median)*(total number of profiles -number of observations in sub-median)*(total number of profiles+1)/12)
In our Example  E_w = sqrt(13*(92-13)*(92+1)/12) = 89
Sum of ranks is somme of 1 to 13 = 91

Step 4: Z =( 91-604.5)/89 = -5.8

**Step 5: Using a Z-test lookup table for a lower tailed test p = 0.05 has a value of -2.576. So we reject the null hypothesis that the first 13 ordered profiles belong to the population of lidar nightly lidar profiles which are distributed about the nightly median.**
* * *
l. 263: As mentioned above, please clearly distinguish between "background" (the average count rate above the usable range, i.e. at 120-153 km) and "noise" (the statistical uncertainty of the count rate at a given height). The background count rate has its noise, but also the signal has. Colloquial both terms are often confused but must not in a publication.

**Changed line 263:**
**"Consider that a poor quality lidar profile which has a signal to noise ratio of 1 at 70 km contributes more information from the signal than from the (background + noise) at 60 km, but information from the (background + noise) than from the signal at 80 km."**

**Changed line 267:**
**The background (noise) of an individual profile, $N_{i}$, is expressed as the summation of photon counts in bins which fall between 120 km and 155 km and the nightly background, $N_{sum}$ is the summation of all $N_{i}$ for the night.**
* * *
Section 3.4: You should make clear that this Section is of different "character" than the other ones. Here you describe some (important) instrumental changes that have been done in the past, not your recent software developments. The removal of SIN is described in Section 3.5.3, i.e. either discard this paragraph in 3.4 or add a reference.

**We take your point that this section is a little different than the surrounding sections. Certainly the hardware changes are not part of the improved algorithm. However, we feel this section fits best here, since it is preceded by a discussion about SNR, and 3.4 explains how we've done our best over 20 years to reduce the noise. The noise discussed here is total background and is not SIN, so it requires its own description here.**
* * *
l. 346: It should read "X to Y km", but the text in brackets can also be removed. The remaining text is clear enough.

**Changed.**
* * *
l. 423: Please explain (in Section 2?) how you combine the N2 Raman channel with the Rayleigh channel. How is the aerosol correction done? Is the Raman channel simply treated as the molecular channel below 30 km?

**The focus of this article is above 30 km.**
**The Raman channel was treated just like a molecular channel. I used a simple model aerosol profile to correct for transmission at both 532 nm and 607 nm. The main benefit of including the N2 Raman channel in this work is to ensure that the count rates in the low gain Rayleigh channel are linear.**
* * *
Fig. 10: The paper mainly deals with nightly mean profiles; therefore I recommend to show a mean profile here. Furthermore you should indicate the transitions between the channels. I generally doubt that an error of 30 % is useful, even for statistical analyses. I recommend using much smaller error margins to avoid unrealistic temperature gradients.

**Figure 10 caption changed to include the word "nightly average".**

**The transition altitude ranges vary from night to night based on the relative signal quality in each lidar channel and the vertical integration. It is a gradual transition, produced by an average weighted by uncertainty in each contributing channel at each altitude. (Alpers et al., 2004) provides a good description of the technique.**

**Readers of the paper will be interested to see the effects of the new algorithm at the higher resolutions presented here, even if the uncertainties are large. It is difficult to find any other long-term dataset for temperatures from 5 to 80 km at 300 m vertical resolution. The presentation here allows readers to see the limits of this method - warts and all. This is not the resolution that we would conduct our geophysics at. Rather it is for the purposes of algorithm testing. The measurements can, of course, be integrated to lower altitude resolution, reducing the uncertainty significantly and smoothing the temperature gradients to more realistic values.**
* * *
l. 452: Please explain in some more detail the differences between the standard NDACC retrieval and yours. I assume that also NDACC has some quality assurance measures like removal of SIN contamination, removal of TES, or deadtime correction.

**We have added to the beginning of the re-worked section 4 (see full Section 4 in response to a later comment):**

**"The NDACC algorithm contains such corrections as deadtime, background, and transmission. The new algorithm improves upon the background correction and identification of bad profiles, and introduces corrections for: signal spikes, TES, identification of good profiles, and noise reduction, all which have not previously been addressed by the NDACC algorithm."**
* * *
Fig. 11: This Figure somewhat demonstrates the confusion about the goals of the paper. The improved retrieval is effective mainly above 75 km. The validation by the co-located lidar is made below 75 km. The companion paper uses data up to 80 km. I do not criticize this figure, but would like to see some clarifications throughout the manuscript, which particular topic is addressed. This would help the reader finding the context of this long paper.

**The re-wording and clarifications in the new Section 4 (see response to other comment below) should help with this matter. Likewise, the description of the goals in the introduction.**

**Up to 70 km, and even 75 km, the improved LTA algorithm has no effect on retrieved temperatures. There, the LTA and LiO3S results match to within their uncertainties even when both lidars use the NDACC algorithm. So below 75 km, the lidars already agree.**

**Above 75 km, the changes as a result of the new LTA algorithm are noticeable. From 70 to 80 km, the LiO3S measurements are still present. Therefore, there is a small region of 5 km in which the LTA NDACC result and the LTA new algorithm result can each be compared to the LiO3S NDACC result. The effects of the new LTA algorithm are to make the LTA and LiO3S come into much closer agreement. Therefore we see this as a success of the new algorithm.**
**Of course, we would prefer to have a co-located lidar which routinely produces temperatures to higher altitudes for comparison.**

**Considering that Fig 11 shows improvements in the LTA retrieval with respect to satellite and model comparisons which account for almost half of the discrepancies seen between LTA NDACC retrievals and the satellites and model, including at altitudes up to 90 km, we are encouraged that the new LTA algorithm is an improvement at higher altitudes as well.**
* * *
Figure 11 (second topic): Is the variance of the SABER data relevant? I assume it shows mainly geophysical variation of the temperatures in the data set. Median errors of all profiles are more

important. How many profiles contribute to the ensemble, what are the criteria for temporal and spatial matching?

**Yes the variance shows the spread of the data due to both errors and geophysical variability.**

**Figure 11 has been updated with all ensemble median errors.**

**N = 212 lidar nights**

**We have updated the text in Section 4 (see full updated text in response to later comment) to say:**
**"This possibility is explored further in the companion paper, and all coincidence criteria for the satellite comparisons are available therein (Wing et al., 2018b)."**
* * *
l. 480: It is confusing that you stress the advantages of your retrieval compared to NDACC in the first part of this paper, but then use the NDACC code for comparison between the lidars.

**We have addressed this confusion by re-writing section 4.**
* * *
l. 514: Before, a resolution of 300 m has been mentioned.
**We have re-written Section 4 to clarify the difference between the vertical resolution of the raw lidar profiles, the algorithm development conducted at 300 m resolution for LTA and the integration of the resulting temperature profiles to the lower 2 km effective resolution for direct comparison with results calculated using the NDACC algorithm.**

**Section 4 now reads:**

**"The NDACC algorithm contains such corrections as deadtime, background, and transmission. The new algorithm improves upon the background correction and identification of bad profiles, and introduces corrections for: signal spikes, TES, identification of good profiles, and noise reduction, all which have not previously been addressed by the NDACC algorithm.**

**The LTA data is collected at 75 m resolution. The spike and TES corrections described in Section 3.3.1 and 3.3.2 are carried out at this resolution. Then the profiles are integrated to 300 m, at which point the remainder of the corrections in Section 3 are applied.**

Temperature profiles using the new algorithm are calculated at 300 m resolution for LTA, and are plotted as the green line in Fig 11. This is higher resolution than the standard NDACC temperature resolution, which is 1 km, smoothed to 2 km effective vertical resolution. The LTA NDACC-calculated temperatures (black line in Fig 11) are plotted at 2 km effective resolution.  By implementing the new algorithm, we have cooled the UMLT lidar temperature retrievals with respect to the standard NDACC temperature algorithm. The modifications cool the mesospheric retrievals by approximately 5 K near 85 km and 20 K by 90 km. There is no significant difference between the new and the NDACC algorithms for LTA below 70 km.

Temperature profiles calculated for LiO3S are all carried out using the NDACC algorithm at an effective vertical resolution of 2 km, and these are show as the orange line in Fig 11. Whereas the LTA NDACC algorithm results are warmer than the LiO3S NDACC algorithm results above 70 km, we now see that the LTA new algorithm results are cooled sufficiently that they more closely match the LiO3S measurements up to 78 km. Therefore the corrections for LTA proposed in the new algorithm represent a significant improvement over the LTA NDACC algorithm for altitudes above 70 km.

A comparison with temperature retrievals from the satellites MLS (red line in Fig 11)  and SABER  (blue with median error and shaded ensemble variance), and with the MSIS-90 model (magenta line in Fig 11),  also show an improvement in the LTA temperatures retrieved using the new algorithm as compared to the LTA NDACC algorithm.  By implementing the techniques described in the sections above we can account for nearly half of the temperature difference between the lidar and the satellites at 90 km. The character change in the difference functions above and below 84 km is in part due to the increasing contributions of the species specific Rayleigh backscattering correction and the corrections to the gravity vector. The remaining temperature difference between the improved lidar temperatures (green) and the satellites and model may be in part due to distortions in the satellite a priori for the geopotential vector. This possibility is explored further in the companion paper, and all coincidence criteria for the satellite comparisons are available therein (Wing et al., 2018b).

It is important to note that additional complications exist when comparing temperatures derived from ground based lidars to temperatures derived from satellite data which have their own calibration concerns. We explore the issues of lidar-satellite comparison in Part B of this paper. A co-located ground-based resonance Doppler or Boltzmann lidar would provide a better comparison data set as resonance lidars have high signal to noise ratios above 75 km (Alpers et al., 2004)."
* * *
l. 516: Do you really mean 602 K?

**Corrected to 6 K**
* * *
l. 543: Here, 300 m resolution is used again. Please explain or unify.

**We have unified to 300 m, and have further explained.**

**Modfied text:**
**For reference, a typical LTA temperature profile with an effective vertical resolution of 300 m (e.g. as shown in Fig 10; the resolution at which we apply the new correction algorithms, which has a detectable effect above 70 km) has an uncertainty due to statistical error of < 1 K below 50 km; 4 K at 60 km; 10 K at 70 km; 50 K at 80 km. LTA temperature profiles are then integrated to 1 km and then smoothed to an effective vertical resolution of 2 km for comparison with the lower-resolution LiO3S, and for submission to the NDACC database. The same typical LTA temperature profile with an effective vertical resolution of 2 km has an uncertainty due to statistical error of 0.2 K at 40 km; 0.4 K at 50 km; 0.6 K at 60 km; 0.7 K at 70 km; 1.8 K at 80 km; and 6 K at 90 km.  A typical LiO3S temperature profile cannot be integrated to the required heights at 300 m resolution due to low signal rates. It is integrated to 1 km, and smoothed to an effective vertical resolution of 2 km, resulting in typical uncertainties due to statistical error of 0.3 K at 40 km; 0.5 K at 50 km; 1.0 K at 60 km; 2.7 K at 70 km; and 10 K at 80 km.**
* * *
There are a lot of typos and odd formulations throughout the manuscript. I recommend consulting advice.

**We have had the paper proofread again following the updates.**

[revised manuscript text omitted]